# OTOv2: Automatic, Generic, User-Friendly

**Tianyi Chen,**[*] **Luming Liang, Tianyu Ding, Ilya Zharkov**
Microsoft
Redmond, WA 98052, USA
{tiachen,lulian,tianyuding,zharkov}@microsoft.com

**Zhihui Zhu**[†]
The Ohio State of University
Columbus, OH 43210, USA
zhu.3440@osu.edu

## ABSTRACT

The existing model compression methods via structured pruning typically require complicated multi-stage procedures. Each individual stage necessitates numerous engineering efforts and domain-knowledge from the end-users which prevent their wider applications onto broader scenarios. We propose the second generation of Only-Train-Once (OTOv2), which *first* automatically trains and compresses a general DNN only once from scratch to produce a more compact model with competitive performance without fine-tuning. OTOv2 is *automatic* and *pluggable* into various deep learning applications, and requires almost *minimal* engineering efforts from the users. Methodologically, OTOv2 proposes two major improvements: *(i)* Autonomy: automatically exploits the dependency of general DNNs, partitions the trainable variables into Zero-Invariant Groups (ZIGs), and constructs the compressed model; and *(ii)* Dual Half-Space Projected Gradient (DHSPG): a novel optimizer to more reliably solve structured-sparsity problems. Numerically, we demonstrate the generality and autonomy of OTOv2 on a variety of model architectures such as VGG, ResNet, CARN, ConvNeXt, DenseNet and StackedUnets, the majority of which cannot be handled by other methods without extensive handcrafting efforts. Together with benchmark datasets including CIFAR10/100, DIV2K, Fashion-MNIST, SVNH and ImageNet, its effectiveness is validated by performing competitively or even better than the state-of-the-arts. The source code is available at https://github.com/tianyic/only_train_once.

## 1 INTRODUCTION

Large-scale Deep Neural Networks (DNNs) have demonstrated successful in a variety of applications (He et al., 2016). However, how to deploy such heavy DNNs onto resource-constrained environments is facing severe challenges. Consequently, in both academy and industry, compressing full DNNs into slimmer ones with negligible performance regression becomes popular. Although this area has been explored in the past decade, it is still far away from being fully solved.

Weight pruning is perhaps the most popular compression method because of its generality and ability in achieving significant reduction of FLOPs and model size by identifying and removing redundant structures (Gale et al., 2019; Han et al., 2015; Lin et al., 2019). However, most existing pruning methods typically proceed a complicated multi-stage procedure as shown in Figure 1, which has apparent limitations: *(i)* **Hand-Craft and User-Hardness**: requires significant engineering efforts and expertise from users to apply the methods onto their own scenarios; *(ii)* **Expensiveness**: conducts DNN training multiple times including the foremost pre-training, the intermediate training for identifying redundancy and the afterwards fine-tuning; and *(iii)* **Low-Generality**: many methods are designed for specific architectures and tasks and need additional efforts to be extended to others.

To address those drawbacks, we naturally need a DNN training and pruning method to achieve the

***Goal.*** *Given a general DNN, automatically train it only once to achieve both high performance and slimmer model architecture simultaneously without pre-training and fine-tuning.*

To realize, the following key problems need to be resolved systematically. *(i)* What are the removal structures (see Section 3.1 for a formal definition) of DNNs? *(ii)* How to identify the redundant

---

[*]Corresponding Author. [†]Partially supported by NSF grant CCF-2240708.

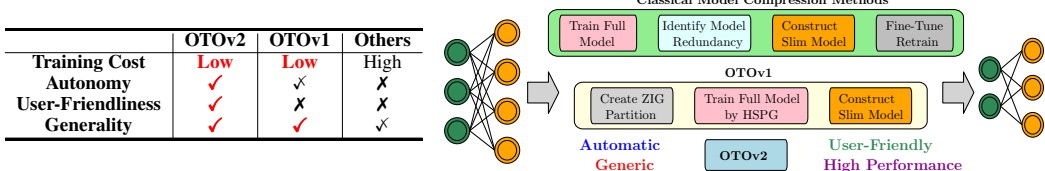

Figure 1: OTOv2 versus existing methods.

removal structures? *(iii)* How to effectively remove redundant structures without deteriorating the model performance to avoid extra fine-tuning? *(iv)* How to make all the above proceeding automatically? Addressing them is challenging in the manner of both algorithmic designs and engineering developments, thereby is not achieved yet by the existing methods to our knowledge.

To resolve *(i-iii)*, Only-Train-Once (OTOv1) (Chen et al., 2021b) proposed a concept so-called Zero-Invariant Group (ZIG), which is a class of minimal removal structures that can be safely removed without affecting the network output if their parameters are zero. To jointly identify redundant ZIGs and achieve satisfactory performance, OTOv1 further proposed a Half-Space Projected Gradient (HSPG) method to compute a solution with both high performance and group sparsity over ZIGs, wherein zero groups correspond to redundant removal structures. As a result, OTOv1 trains a full DNN from scratch only once to compute a slimmer counterpart exhibiting competitive performance without fine-tuning, and is perhaps the closest to the goal among the existing competitors.

Nevertheless, the fundamental problem *(iv)* is not addressed in OTOv1, *i.e.*, the ZIGs partition is not automated and only implemented onto several specific architectures. OTOv1 suffers a lot from requiring extensive hand-crafting efforts and domain knowledge to partition trainable variables into ZIGs which prohibits its broader usages. Meanwhile, OTOv1 highly depends on HSPG to yield a solution with both satisfactory performance and high group sparsity. However, the sparsity exploration of HSPG is typically sensitive to the regularization coefficient thereby requires time-consuming hyper-parameter tuning and lacks capacity to precisely control the ultimate sparsity level.

To overcome the drawbacks of OTOv1 and simultaneously tackle *(i-iv)*, we propose Only-Train-Once v2 (OTOv2), the next-generation one-shot deep neural network training and pruning framework. Given a full DNN , OTOv2 is able to train and compress it from scratch into a slimmer DNN with significant FLOPs and parameter quantity reduction. In contrast to others, OTOv2 drastically simplifies the complicated multi-

```
                        ── Library Usage ──
1  from only_train_once import OTO
2  # General DNN model
3  oto = OTO(model)
4  optimizer = oto.dhspg()
5  # Train as normal
6  optimizer.step()
7  oto.compress()
```

stage procedures; **guarantees performance** more reliably than OTOv1; and is **generic**, **automatic** and **user-friendly**. Our main contributions are summarized as follows.

- **Infrastructure for Automated DNN One-Shot Training and Compression.** We propose and develop perhaps the first generic and automated framework to compress a general DNN with both excellent performance and substantial complexity reduction in terms of FLOPs and model cardinality. OTOv2 only trains the DNN once, neither pre-training nor fine-tuning is a necessity. OTOv2 is user-friendly and easily applied onto generic tasks as shown in **library usage**. Its success relies on the breakthroughs from both algorithmic designs and infrastructure developments.

- **Automated ZIG Partition and Automated Compressed Model Construction.** We propose a novel graph algorithm to automatically exploit and partition the variables of a general DNN into Zero-Invariant Groups (ZIGs), *i.e.*, the minimal groups of parameters that need to be pruned together. We further propose a novel algorithm to automatically construct the compressed model by the hierarchy of DNN and eliminating the structures corresponding to ZIGs as zero. Both algorithms are dedicately designed, and work effectively with low time and space complexity.

- **Novel Structured-Sparsity Optimization Algorithm.** We propose a novel optimization algorithm, called Dual Half-Space Projected Gradient (DHSPG), to train a general DNN once from scratch to effectively achieve competitive performance and high group sparsity in the manner of ZIGs, which solution is further leveraged into the above automated compression. DHSPG formulizes a constrained sparse optimization problem and solves it by constituting a direction within the intersection of dual half-spaces to largely ensure the progress to both the objective convergence and the identification of redundant groups. DHSPG outperforms the HSPG in OTOv1 in terms of enlarging search space, fewer hyper-parameter tuning, and more reliably controlling sparsity.

- **Experimental Results.** We apply OTOv2 onto a variety of DNNs (most of which have structures with complicated connectivity) and extensive benchmark datasets, including CIFAR10/100, DIV2K, SVNH, Fashion-MNIST and ImageNet. OTOv2 trains and compresses various DNNs simultaneously from scratch without fine-tuning for significant inference speedup and parameter reduction, and achieves competitive or even state-of-the-art results on compression benchmarks.

## 2  RELATED WORK

**Structured Pruning.** To compute compact architectures for efficient model inference and storage, structured pruning identifies and prunes the redundant structures in a full model (Gale et al., 2019; Han et al., 2015). The general procedure can be largely summarized as: *(i)* train a full model; *(ii)* identify and remove the redundant structures to construct a slimmer DNN based on various criteria, including (structured) sparsity (Lin et al., 2019; Wen et al., 2016; Li et al., 2020b; Zhuang et al., 2020; Chen et al., 2017; 2018; 2021a; 2020a; Gao et al., 2020; Zhuang et al., 2020; Meng et al., 2020; Yang et al., 2019), Bayesian pruning (Zhou et al., 2019; Louizos et al., 2017; van Baalen et al., 2020), ranking importance (Li et al., 2020a; Luo et al., 2017; Hu et al., 2016; He et al., 2018a; Li et al., 2019; Zhang et al., 2018), reinforcement learning (He et al., 2018b; Chen et al., 2019), lottery ticket (Frankle & Carbin, 2018; Frankle et al., 2019; Renda et al., 2020), etc.; *(iii)* (iteratively) retrain the pruned model to regain the accuracy regression during pruning. These methods have to conduct a complicated and time-consuming procedure to trains the DNN multiple times and requires a good deal of domain knowledge to manually proceed every individual step. OTOv1 (Chen et al., 2021b) is recently proposed to avoid fine-tuning and end-to-end train and compress the DNN once, whereas its automation relies on spending numerous handcrafting efforts on creating ZIGs partition and slimmer model construction for specific target DNNs in advance, thereby is actually semi-automated.

**Automated Machine Learning (AutoML).** OTOv2 fills into a vital gap within AutoML domain regarding given an general DNN architecture, how to automatically train and compress it into a slimmer one with competitive performance and significant FLOPs and parameter quantity reduction. The existing AutoML methods focus on *(i)* automated feature engineering (Kanter & Veeramachaneni, 2015), *(ii)* automated hyper-parameter setting (Klein et al., 2017), and *(iii)* neural architecture search (NAS) (Elsken et al., 2018). NAS searches a DNN architecture with satisfactory performance from a prescribed fixed full graph wherein the connection between two nodes (tensors) is searched from a pool of prescribed operators. NAS itself has no capability to slim and remove redundancy from the searched architectures due to the pool being fixed and is typically time-consuming. As a result, NAS may serve as a prerequisite step to search a target network architecture as the input to OTOv2.

## 3  OTOv2

OTOv2 has nearly reached the *goal* of model compression via weight pruning, which is outlined in Algorithm 1. In general, given a neural network $\mathcal{M}$ to be trained and compressed, OTOv2 first automatically figures out the dependencies among the vertices to exploit minimal removal structures and partitions the trainable variables into Zero-Invariant Groups (ZIGs) (Algorithm 2). ZIGs ($\mathcal{G}$) are then fed into a structured sparsity optimization problem, which is solved by a Dual Half-Space Projected Gradient (DHSPG) method to yield a solution $x^*_{\text{DHSPG}}$ with competitive performance as well as high group sparsity in the view of ZIGs (Algorithm 3). The compressed model $\mathcal{M}^*$ is ultimately constructed via removing the redundant structures corresponding to the ZIGs being zero. $\mathcal{M}^*$ significantly accelerates the inference in both time and space complexities and returns the identical outputs to the full model $\mathcal{M}$ parameterized as $x^*_{\text{DHSPG}}$ due to the properties of ZIGs, thus avoids further fine-tuning $\mathcal{M}^*$. The whole procedure is proceeded automatically and easily employed onto various DNN applications and consumes almost minimal engineering efforts from the users.

---

**Algorithm 1** Outline of OTOv2.

---

1: **Input.** An arbitrary full model $\mathcal{M}$ to be trained and compressed (no need to be pretrained).
2: **Automated ZIG Partition.** Partition the trainable parameters of $\mathcal{M}$ into $\mathcal{G}$.
3: **Train $\mathcal{M}$ by DHSPG.** Seek a highly group-sparse solution $x^*_{\text{DHSPG}}$ with high performance.
4: **Automated Compressed Model $\mathcal{M}^*$ Construction.** Construct a slimmer model upon $x^*_{\text{DHSPG}}$.
5: **Output:** Compressed slimmed model $\mathcal{M}^*$.

---

## 3.1 AUTOMATED ZIG PARTITION

**Background.** We review relevant concepts before describing how to proceed ZIG partition automatically. Due to the complicated connectivity of DNNs, removing an arbitrary structure or component may result in an invalid DNN. We say a structure *removal* if and only if the DNN without this component still serves as a valid DNN. Consequently, a removal structure is called *minimal* if and only if it can not be further decomposed into multiple removal structures. A particular class of minimal removal structures—that produces zero outputs to the following layer if their parameters being zero—are called ZIGs (Chen et al., 2021b) which can be removed directly without affecting the network output. Thus, each ZIG consists of a minimal group of variables that need to be pruned together and dominates most DNN structures, *e.g.*, layers as `Conv`, `Linear` and `MultiHeadAtten`. While ZIGs exist for general DNNs, their topology can vary significantly due to the complicated connectivity. This together with the lack of API poses severe challenges to automatically exploit ZIGs in terms of both algorithmic designs and engineering developments.

---

**Algorithm 2** Automated Zero-Invariant Group Partition.

---

1: **Input:** A DNN $\mathcal{M}$ to be trained and compressed.
2: Construct the trace graph $(\mathcal{E}, \mathcal{V})$ of $\mathcal{M}$.
3: Find connected components $\mathcal{C}$ over all accessory, shape-dependent joint and unknown vertices.
4: Grow $\mathcal{C}$ till incoming nodes are either stem or shape-independent joint vertices.
5: Merge connected components in $\mathcal{C}$ if any intersection.
6: Group pairwise parameters of stem vertices in the same connected component associated with parameters from affiliated accessory vertices if any as one ZIG into $\mathcal{G}$.
7: **Return** the zero-invariant groups $\mathcal{G}$.

---

**Algorithmic Outline.** To resolve the autonomy of ZIG partition, we present a novel, effective and efficient algorithm. As outlined in Algorithm 2, the algorithm essentially partitions the graph of DNN into a set of connected components of dependency, then groups the variables based on the affiliations among the connected components. For more intuitive illustration, we provide a small but complicated DemoNet along with explanations about its ground truth minimal removal structures (ZIGs) in Figure 2. We now elaborate Algorithm 2 to automatically recover the ground truth ZIGs.

**Graph Construction.** In particular, we first establish the trace graph $(\mathcal{E}, \mathcal{V})$ of the target DNN, wherein each vertex in $\mathcal{V}$ refers to a specific operator, and the edges in $\mathcal{E}$ describe how they connect (line 2 of Algorithm 2). We categorize the vertices into stem, joint, accessory or unknown. Stem vertices equip with trainable parameters and have capacity to transform their input tensors into other shapes, *e.g.*, `Conv` and `Linear`. Joint vertices aggregate multiple input tensors into a single output such as `Add`, `Mul` and `Concat`. Accessory vertices operate a single input tensor into a single output and may possess trainable parameters such as `BatchNorm` and `ReLu`. The remaining unknown vertices proceed some uncertain operations. Apparently, stem vertices compose most of the DNN parameters. Joint vertices establish the connections cross different vertices, and thus dramatically bring hierarchy and intricacy of DNN. To keep the validness of the joint vertices, the minimal removal structures should be carefully constructed. Furthermore, we call joint vertices being input shape dependent (SD) if requiring inputs in the same shapes such as `Add`, otherwise being shape-independent (SID) such as `Concat` along the channel dimension for `Conv` layers as input.

**Construct Connected Components of Dependency.** Now, we need to figure out the exhibiting dependency across the vertices to seek the minimal removal structures of the target DNN. To proceed, we first connect accessory, SD joint and unknown vertices together if adjacent to form a set of connected components $\mathcal{C}$ (see Figure 2c and line 3 of Algorithm 2). This step is to establish the skeletons for finding vertices that depend on each other when considering removing hidden structures. The underlying intuitions of this step in depth are *(i)* the adjacent accessory vertices operate and are subject to the same ancestral stem vertices if any; *(ii)* SD joint vertices force their ancestral stem vertices dependent on each other to yield tensors in the same shapes; and *(iii)* unknown vertices introduce uncertainty, hence finding potential affected vertices is necessary. We then grow $\mathcal{C}$ till all their incoming vertices are either stem or SID joint vertices and merge the connected components if any intersection as line 4-5. Remark here that the newly added stem vertices are affiliated by the accessory vertices, such as `Conv1` for `BN1-ReLu` and `Conv3+Conv2` for `BN2|BN3` in Figure 2d. In addition, the SID joint vertices introduce dependency between their affiliated accessory vertices and incoming connected components, *e.g.*, `Concat-BN4` depends on both `Conv1-BN1-ReLu` and `Conv3+Conv2-BN2|BN3` since `BN4` normalizes their concatenated tensors along channel.

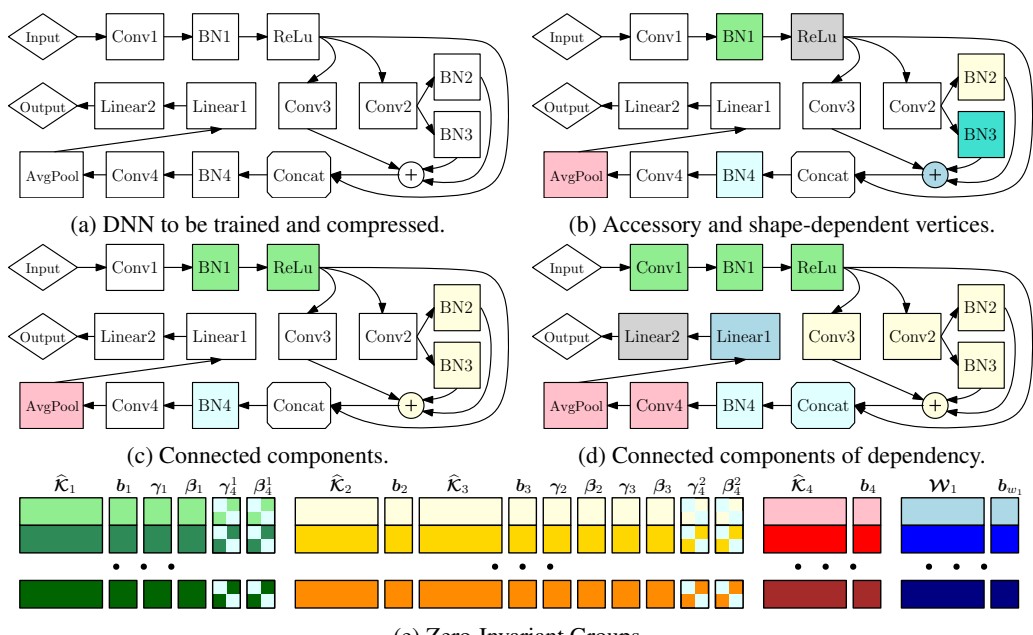

(a) DNN to be trained and compressed.

(b) Accessory and shape-dependent vertices.

(c) Connected components.

(d) Connected components of dependency.

(e) Zero-Invariant Groups.

Figure 2: Automated ZIG partition illustration. $\widehat{\mathcal{K}}_i$ and $\boldsymbol{b}_i$ are the flatten filter matrix and bias vector of `Conv`$i$, where the $j$th row of $\widehat{\mathcal{K}}_i$ represents the $j$th 3D filter. $\boldsymbol{\gamma}_i$ and $\boldsymbol{\beta}_i$ are the weighting and bias vectors of `BN`$i$. $\mathcal{W}_i$ and $\boldsymbol{b}_{w_i}$ are the weighting matrix and bias vector for `Linear`$i$. The ground truth ZIGs $\mathcal{G}$ are present in Figure 2e. Since the output tensors of `Conv2` and `Conv3` are added together, both layers associated with the subsequent `BN2` and `BN3` must remove the same number of filters from $\widehat{\mathcal{K}}_2$ and $\widehat{\mathcal{K}}_3$ and scalars from $\boldsymbol{b}_2, \boldsymbol{b}_3, \boldsymbol{\gamma}_2, \boldsymbol{\gamma}_3, \boldsymbol{\beta}_2$ and $\boldsymbol{\beta}_3$ to keep the addition valid. Since `BN4` normalizes the concatenated outputs along channel from `Conv1-BN1-ReLu` and `Conv3+Conv2-BN2|BN3`, the corresponding scalars in $\boldsymbol{\gamma}_4, \boldsymbol{\beta}_4$ need to be removed simultaneously.

**Form ZIGs.** Finally, we form ZIGs based on the connected components of dependency as Figure 2d. The pairwise trainable parameters across all individual stem vertices in the same connected component need to be first grouped together as Figure 2e, wherein the parameters of the same color represent one group. Later on, the accessory vertices insert their trainable parameters if applicable into the groups of their dependent stem vertices accordingly. Some accessory vertex such as `BN4` may depend on multiple groups because of the SID joint vertex, thereby its trainable parameters $\boldsymbol{\gamma}_4$ and $\boldsymbol{\beta}_4$ need to be partitioned and separately added into corresponding groups, *e.g.*, $\boldsymbol{\gamma}_4^1, \boldsymbol{\beta}_4^1$ and $\boldsymbol{\gamma}_4^2, \boldsymbol{\beta}_4^2$. In addition, the connected components that are adjacent to the output of DNN are excluded from forming ZIGs since the output shape should be fixed such as `Linear2`. For safety, the connected components that possess unknown vertices are excluded as well due to uncertainty, which further guarantees the generality of the framework applying onto DNNs with customized operators.

**Complexity Analysis.** The proposed automated ZIG partition Algorithm 2 is a series of customized graph algorithms dedicately composed together. In depth, every individual sub-algorithm is achieved by depth-first-search recursively traversing the trace graph of DNN and conducting step-specific operations, which has time complexity as $\mathcal{O}(|\mathcal{V}| + |\mathcal{E}|)$ and space complexity as $\mathcal{O}(|\mathcal{V}|)$ in the worst case. The former one is computed by discovering all neighbors of each vertex by traversing the adjacency list once in linear time. The latter one is because the trace graph of DNN is acyclic thereby the memory cache consumption is up to the length of possible longest path for an acyclic graph as $|\mathcal{V}|$. Therefore, automated ZIG partition can be efficiently completed within linear time.

## 3.2 DUAL HALF-SPACE PROJECTED GRADIENT (DHSPG)

Given the constructed ZIGs $\mathcal{G}$ by Algorithm 2, the next step is to jointly identify which groups are redundant to be removed and train the remaining groups to achieve high performance. To tackle it, we construct a structured sparsity optimization problem and solve it via a novel DHSPG. Compared with HSPG, DHSPG constitutes a dual-half-space direction with automatically selected regularization coefficients to more reliably control the sparsity exploration, and enlarges the search space by partitioning the ZIGs into separate sets to avoid trapping around the origin for better generalization.

**Target Problem.** Structured sparsity inducing optimization problem is a natural choice to seek a group sparse solution with high performance, wherein the zero groups refer to the redundant structures, and the non-zero groups exhibit the prediction power to maintain competitive performance to the full model. We formulate an optimization problem with a group sparsity constraint in the form of ZIGs $\mathcal{G}$ as (1) and propose a novel Dual Half-Space Projected Gradient (DHSPG) to solve it.

$$\underset{\boldsymbol{x} \in \mathbb{R}^n}{\text{minimize}} \, f(\boldsymbol{x}), \quad \text{s.t. } \text{Card}\{g \in \mathcal{G} | [\boldsymbol{x}]_g = 0\} = K, \tag{1}$$

where $K$ is the target group sparsity level. Larger $K$ indicates higher group sparsity in the solution and typically results in more aggressive FLOPs and parameter quantity reductions.

**Related Optimizers and Limitations.** To solve such constrained problem, ADMM converts it into a min-max problem, but can not tackle the non-smooth and non-convex hard constraint of sparsity without hurting the objective, thus necessitates extra fine-tuning afterwards (Lin et al., 2019). HSPG in OTOv1 (Chen et al., 2021b) and proximal methods (Xiao & Zhang, 2014) relax it into a non-constrained mixed $\ell_1/\ell_p$ regularization problem, but can not guarantee the sparsity constraint because of the implicit relationship between the regularization coefficient and the sparsity level. In addition, the augmented regularizer penalizes the magnitude of the entire trainable variables which restricts the search space to converge to the local optima nearby the origin point, *e.g.*, $\boldsymbol{x}_1^*$ in Figure 3. However, the local optima with the highest generalization may locate variably for different applications, and some may stay away from the origin point, *e.g.*, $\boldsymbol{x}_2^*, \cdots, \boldsymbol{x}_5^*$ in Figure 3.

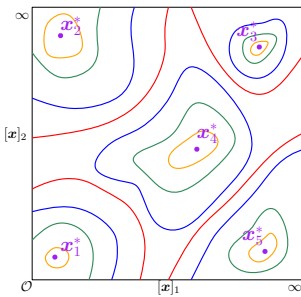

Figure 3: Local optima $\boldsymbol{x}^* \in \mathbb{R}^2$ distribution over the objective landscape.

**Algorithm Outline for DHSPG.** To resolve the drawbacks of the existing optimization algorithms for solving (1), we propose a novel algorithm, named Dual Half-Space Projected Gradient (DHSPG), stated as Algorithm 3, with two takeaways.

**Partition Groups.** To avoid always trapping in the local optima near the origin point, we further partition the groups in $\mathcal{G}$ into two subsets: one has magnitudes of variables being penalized $\mathcal{G}_p$, and the other does not force to penalize variable magnitude $\mathcal{G}_{np}$. Different criteria can be applied here to construct the above partition based on salience scores, *e.g.*, cosine-similarity $\cos(\theta_g)$ between the pro-

---

**Algorithm 3** Dual Half-Space Projected Gradient (DHSPG)

1: **Input:** initial variable $\boldsymbol{x}_0 \in \mathbb{R}^n$, initial learning rate $\alpha_0$, warm-up steps $T_w$, half-space project steps $T_h$, target group sparsity $K$ and ZIGs $\mathcal{G}$.
2: Warm up $T_w$ steps via stochastic gradient descent.
3: Construct $\mathcal{G}_p$ and $\mathcal{G}_{np}$ given $\mathcal{G}$ and $K$ as (2).
4: **for** $t = T_w, T_w + 1, T_w + 2, \cdots,$ **do**
5:     Compute gradient estimate $\nabla f(\boldsymbol{x}_t)$ or its variant.
6:     Update $[\boldsymbol{x}_{t+1}]_{\mathcal{G}_{np}}$ as $[\boldsymbol{x}_t - \alpha_t \nabla f(\boldsymbol{x}_t)]_{\mathcal{G}_{np}}$.
7:     Select proper $\lambda_g$ for $g \in \mathcal{G}_p$.
8:     Compute $[\tilde{\boldsymbol{x}}_{t+1}]_{\mathcal{G}_p}$ via subgradient descent of $\psi$.
9:     **if** $t \geq T_h$ **then**
10:         Perform Half-Space projection over $[\tilde{\boldsymbol{x}}_{t+1}]_{\mathcal{G}_p}$.
11:     Update $[\boldsymbol{x}_{t+1}]_{\mathcal{G}_p} \leftarrow [\tilde{\boldsymbol{x}}_{t+1}]_{\mathcal{G}_p}$.
12:     Update $\alpha_{t+1}$.
13: **Return** the final iterate $\boldsymbol{x}_{\text{DHSPG}}^*$.

---

jection direction $-[\boldsymbol{x}]_g$ and the negative gradient or its estimation $-[\nabla f(\boldsymbol{x})]_g$. Higher cos-similarity over $g \in \mathcal{G}$ indicates that projecting the group of variables in $g$ onto zeros is more likely to make progress to the optimality of $f$ (considering the descent direction from the perspective of optimization). The magnitude over $[\boldsymbol{x}]_g$ then needs to be penalized. Therefore, we compute $\mathcal{G}_p$ by picking up the ZIGs with top-$K$ highest salience scores and $\mathcal{G}_{np}$ as its complementary as (2). To compute more reliable scores, the partition is proceeded after performing $T_w$ warm-up steps as line 2-3.

$$\mathcal{G}_p = (\text{Top-}K) \underset{g \in \mathcal{G}}{\arg\max} \, \text{salience-score}(g) \text{ and } \mathcal{G}_{np} = \{1, 2, \cdots, n\} \backslash \mathcal{G}_p. \tag{2}$$

**Update Variables.** For the variables in $\mathcal{G}_{np}$ of which magnitudes are not penalized, we proceed vanilla stochastic gradient descent or its variants, such as Adam (Kingma & Ba, 2014), *i.e.*, $[\boldsymbol{x}_{t+1}]_{\mathcal{G}_{np}} \leftarrow [\boldsymbol{x}_t]_{\mathcal{G}_{np}} - \alpha_t [\nabla f(\boldsymbol{x}_t)]_{\mathcal{G}_{np}}$. For the groups of variables in $\mathcal{G}_p$ to penalize magnitude, we seek to find out redundant groups as zero, but instead of directly projecting them onto zero as ADMM which easily destroys the progress to the optimum, we formulate a relaxed non-constrained subproblem as (3) to gradually reduce the magnitudes without deteriorating the objective and project groups onto zeros if the projection serves as a descent direction during the training process.

$$\underset{[\boldsymbol{x}]_{\mathcal{G}_p}}{\text{minimize}}\ \psi([\boldsymbol{x}]_{\mathcal{G}_p}) := f\left([\boldsymbol{x}]_{\mathcal{G}_p}\right) + \sum_{g \in \mathcal{G}_p} \lambda_g \left\|[\boldsymbol{x}]_g\right\|_2, \tag{3}$$

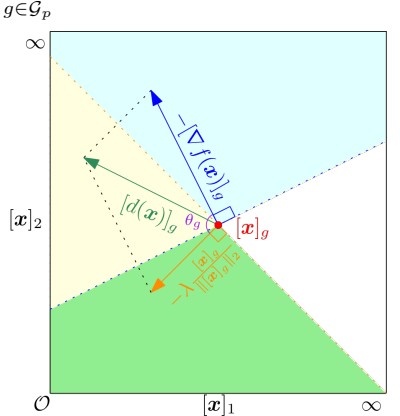

where $\lambda_g$ is a group-specific regularization coefficient and needs to be dedicatedly chosen to guarantee the decrease of both the variable magnitude for $g$ as well as the objective $f$. In particular, we compute a negative subgradient of $\psi$ as the search direction $[\boldsymbol{d}(\boldsymbol{x})]_{\mathcal{G}_p} := -[\nabla f(\boldsymbol{x})]_{\mathcal{G}_p} - \sum_{g \in \mathcal{G}_p} \lambda_g [\boldsymbol{x}]_g / \max\{\|[\boldsymbol{x}]_g\|_2, \tau\}$ with $\tau$ as a safeguard constant. To ensure $[\boldsymbol{d}(\boldsymbol{x})]_{\mathcal{G}_p}$ as a descent direction for both $f$ and $\|\boldsymbol{x}\|_2$, $[\boldsymbol{d}(\boldsymbol{x})]_g$ needs to fall into the intersection between the *dual half-spaces* with normal directions as $-[\nabla f]_g$ and $-[\boldsymbol{x}]_g$ for any $g \in \mathcal{G}_p$ as shown in Figure 4. In other words, $[\boldsymbol{d}(\boldsymbol{x})]_{\mathcal{G}_p}^{\top}[-\nabla f(\boldsymbol{x})]_{\mathcal{G}_p}$ and $[\boldsymbol{d}(\boldsymbol{x})]_{\mathcal{G}_p}^{\top}[-\boldsymbol{x}]_{\mathcal{G}_p}$ are greater than 0. It further indicates that $\lambda_g$ locates in the interval $(\lambda_{\min,g}, \lambda_{\max,g}) := \left(-\cos(\theta_g)\|[\nabla f(\boldsymbol{x})]_g\|_2, -\frac{\|[\nabla f(\boldsymbol{x})]_g\|_2}{\cos(\theta_g)}\right)$ if $\cos(\theta_g) < 0$

Figure 4: Search direction in DHSPG.

otherwise can be an arbitrary positive constant. Such $\lambda_g$ brings the decrease of both the objective and the variable magnitude. We then compute a trial iterate $[\tilde{\boldsymbol{x}}_{t+1}]_{\mathcal{G}_p} \leftarrow [\boldsymbol{x}_t - \alpha_t \boldsymbol{d}(\boldsymbol{x}_t)]_{\mathcal{G}_p}$ via the subgradient descent of $\psi$ as line 8. The trial iterate is fed into the Half-Space projector (Chen et al., 2021b) which outperforms proximal operators to yield group sparsity more productively without hurting the objective as line 9-10. Remark here that OTOv1 utilizes a global coefficient $\lambda$ for all groups, thus lacks sufficient capability to guarantee both aspects for each individual group in accordance.

**Convergence and Complexity Analysis.** DHSPG converges to the solution of (1) $\boldsymbol{x}^*_{\text{DHSPG}}$ in the manner of both theory and practice. In fact, the theoretical convergence relies on the the construction of dual half-space mechanisms which yield sufficient decrease for both objective $f$ and variable magnitude, see Lemma 2 and Corollary 1 in Appendix C. Together with the sparsity recovery of Half-Space projector (Chen et al., 2020b, Theorem 2), DHSPG effectively computes a solution with desired group sparsity. In addition, DHSPG consumes the same time complexity $\mathcal{O}(n)$ as other first-order methods, such as SGD and Adam, since all operations can be finished within linear time.

### 3.3 AUTOMATED COMPRESSED MODEL CONSTRUCTION

In the end, given the solution $\boldsymbol{x}^*_{\text{DHSPG}}$ with both high performance and group sparsity, we now automatically construct a compact model which is a *manual* step with unavoidable substantial engineering efforts in OTOv1. In general, we traverse all vertices with trainable parameters, then remove the structures in accordance with ZIGs being zero, such as the dotted rows of $\widehat{\mathcal{K}}_1, \widehat{\mathcal{K}}_2, \widehat{\mathcal{K}}_3$ and scalars of $\boldsymbol{b}_2, \boldsymbol{\gamma}_1, \boldsymbol{\beta}_1$ as illustrated in Figure 5. Next, we erase the redundant parameters that affiliate with the

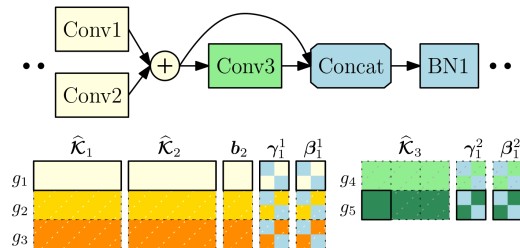

Figure 5: Automated compressed model construction. $\mathcal{G} = \{g_1, g_2, \cdots, g_5\}$ and $[\boldsymbol{x}^*_{\text{DHSPG}}]_{g_2 \cup g_3 \cup g_4} = \boldsymbol{0}$.

removed structures of their incoming stem vertices to keep the operations valid, *e.g.*, the second and third channels in $g_5$ are removed though $g_5$ is not zero. The automated algorithm is promptly complete in linear time via performing two passes of depth-first-search and manipulating parameters to produce a more compact model $\mathcal{M}^*$. Based on the property of ZIGs, $\mathcal{M}^*$ returns the same inference outputs as the full $\mathcal{M}$ parameterized as $\boldsymbol{x}^*_{\text{DHSPG}}$ thus no further fine-tuning is necessary.

## 4 NUMERICAL EXPERIMENTS

We develop OTOv2 to train and compress DNNs into slimmer networks with significant inference speedup and storage saving without fine-tuning. The implementation details are presented in Appendix A. To demonstrate its effectiveness, we first verify the correctness of automated ZIG partition and automated compact model construction by employing OTOv2 onto a variety of DNNs with

complicated structures (see the visualizations in Appendix D). Then, we compare OTOv2 with other methods on the benchmark experiments to show its competitive (or even superior) performance. In addition, we conduct ablation studies of DHSPG versus HSPG on the popular super-resolution task and Bert (Vaswani et al., 2017) on Squad (Rajpurkar et al., 2016) in Appendix B. Together with autonomy, user-friendliness and generality, OTOv2 arguably becomes the new state-of-the-art.

**Sanity of Automated ZIG and Automated Compression.** The foremost step is to validate the correctness of the whole framework including both algorithm designs and infrastructure developments. We select five DNNs with complex topological structures, *i.e.*, StackedUnets, DenseNet (Huang et al., 2017), ConvNeXt (Liu et al., 2022) and CARN (Ahn et al., 2018) (see Appendix B for details), as well as DemoNet in Section 3.1, all of which are not easily to be compressed via the existing non-automatic methods unless with sufficient domain knowledge and extensive handcrafting efforts. Remark here that StackedUnets consumes two input tensors, and is constructed by stacking two standard Unets (Ronneberger et al., 2015) with different downsamplers and aggregating the corresponding two outputs together. To intuitively illustrate the automated ZIG partition over these complicated structures, we provide the visualizations of the connected components of dependency in Appendix D. To quantitatively measure the performance of OTOv2, we further employ these model architectures onto a variety of benchmark datasets, *e.g.*, Fashion-MNIST (Xiao et al., 2017), SVNH (Netzer et al., 2011), CIFAR10/100 (Krizhevsky & Hinton, 2009) and ImageNet (Deng et al., 2009). The main results are presented in Table 1.

Compared with the baselines trained by vanilla SGD, under the same amount of training cost, OTOv2 automatically reaches not only the competitive performance but also remarkable speed up in terms of FLOPs and parameter quantity reductions. In particular, the slimmer DemoNet and

Table 1: OTOv2 on extensive DNNs and datasets.

| Backend | Dataset | Method | FLOPs | # of Params | Top-1 Acc. |
|---|---|---|---|---|---|
| DemoNet | Fashion-MNIST | Baseline | 100% | 100% | 84.5% |
| DemoNet | Fashion-MNIST | **OTOv2** | **24.0%** | **23.3%** | **84.3%** |
| StackedUnets | SVNH | Baseline | 100% | 100% | 94.8% |
| StackedUnets | SVNH | **OTOv2** | **26.4%** | **17.0%** | **94.7%** |
| DenseNet121 | CIFAR100 | Baseline | 100% | 100% | 77.0% |
| DenseNet121 | CIFAR100 | **OTOv2** | **20.8%** | **26.7%** | **75.5%** |
| ConvNeXt-Tiny | ImageNet | Baseline | 100% | 100% | 82.0% |
| ConvNeXt-Tiny | ImageNet | **OTOv2** | **52.8%** | **54.2%** | **81.1%** |

StackedUnets computed by OTOv2 negligibly regress the top-1 accuracy by 0.1%-0.2% but significantly reduce the FLOPs and the number of parameters by 73.6%-83.0%. Consistent phenomena also hold for DenseNet121 where the slimmer architecture is about 5 times more efficient than the full models but with competitive accuracy. OTOv2 works with TIMM (Wightman, 2019) to effectively compress ConvNeXt-Tiny which shows its flexibility to the modernized training tricks. The success of OTOv2 on these architectures well validates the sanity of the framework.

**Benchmark Experiments.** The secondary step is to demonstrate the effectiveness of OTOv2 by comparing the performance with other state-of-the-arts on benchmark compression experiments, *i.e.*, common architectures such as VGG16 (Simonyan & Zisserman, 2014) and ResNet50 (He et al., 2016) as well as datasets CIFAR10 (Krizhevsky & Hinton, 2009) and ImageNet (Deng et al., 2009).

**VGG16 on CIFAR10.** We first consider vanilla VGG16 and a variant referred as VGG16-BN that appends a batch normalization layer after every convolutional layer. OTOv2 automatically exploits the minimal removal structures of VGG16 and partitions the trainable variables into ZIGs (see Figure 14 in Appendix D). DHSPG is then triggered over the partitioned ZIGs to train the model from scratch to find a solution with high group sparsity. Finally, a slimmer VGG16 is automatically constructed without any fine-tuning. As shown in Table 2, the slimmer VGG16 leverages only 2.5% of parameters to dramatically reduce the FLOPs by 86.6% with the competitive top-1 accuracy to the full model and other state-of-the-art methods. Likewise, OTOv2 compresses VGG16-BN to maintain the baseline accuracy by the fewest 4.9% of parameters and 23.7% of FLOPs. Though SCP and RP reach higher accuracy, they require significantly 43%-102% more FLOPs than that of OTOv2.

**ResNet50 on CIFAR10.** We now conduct experiments to compare with a few representative automatic pruning methods such as AMC and ANNC. AMC establishes a reinforcement learning agent to guide a layer-wise compression, while it only achieves autonomy over a few prescribed specific models and requires multiple-stage training costs. Simple pruning methods

Table 3: ResNet50 for CIFAR10.

| Method | FLOPs | # of Params | Top-1 Acc. |
|---|---|---|---|
| Baseline | 100% | 100% | 93.5% |
| AMC (He et al., 2018b) | – | 60.0% | 93.6% |
| ANNC (Yang et al., 2020) | – | 50.0% | **95.0%** |
| PruneTrain (Lym et al., 2019) | 30.0% | – | 93.1% |
| N2NSkip (Sharma et al., 2020) | – | 10.0% | 94.4% |
| OTOv1 (Chen et al., 2021b) | 12.8% | 8.8% | 94.4% |
| **OTOv2** (90% group sparsity) | **2.2%** | **1.2%** | 93.0% |
| **OTOv2** (80% group sparsity) | 7.8% | 4.1% | 94.5% |

Table 2: VGG16 and VGG16-BN for CIFAR10. Convolutional layers are in bold.

| Method | BN | Architecture | FLOPs | # of Params | Top-1 Acc. |
|---|---|---|---|---|---|
| Baseline | ✗ | **64-64-128-128-256-256-256-512-512-512-512-512-512**-512-512 | 100% | 100% | 91.6% |
| SBP (Neklyudov et al., 2017) | ✗ | **47-50-91-115-227-160-50-72-51-12-34-39-20**-20-272 | 31.1% | 5.9% | **91.0%** |
| BC (Louizos et al., 2017) | ✗ | **51-62-125-128-228-129-38-13-9-6-5-6-6**-6-20 | 38.5% | 5.4% | **91.0%** |
| RBC (Zhou et al., 2019) | ✗ | **43-62-120-120-182-113-40-12-20-11-6-9-10**-10-22 | 32.3% | 3.9% | 90.5% |
| RBP (Zhou et al., 2019) | ✗ | **50-63-123-108-104-57-23-14-9-8-6-7-11**-11-12 | 28.6% | 2.6% | **91.0%** |
| OTOv1 (Chen et al., 2021b) | ✗ | **21-45-82-110-109-68-37-13-9-7-3-5-8**-170-344 | 16.3% | **2.5%** | **91.0%** |
| **OTOv2** (85% group sparsity) | ✗ | **22-30-56-102-142-101-28-11-6-6-5-5**-101-127 | **13.4%** | **2.5%** | **91.0%** |
| Baseline | ✓ | **64-64-128-128-256-256-256-512-512-512-512-512-512**-512-512 | 100% | 100% | 93.2% |
| EC (Li et al., 2016) | ✓ | **32-64-128-128-256-256-256-256-256-256-256-256**-512-512 | 65.8% | 37.0% | 93.1% |
| Hinge (Li et al., 2020b) | ✓ | – | 60.9% | 20.0% | 93.6% |
| SCP (Kang & Han, 2020) | ✓ | – | 33.8% | 7.0% | 93.8% |
| OTOv1 (Chen et al., 2021b) | ✓ | **22-56-93-123-182-125-95-45-27-21-10-13-19**-244-392 | 26.8% | 5.5% | 93.3% |
| RP (Li et al., 2022) | ✓ | – | 47.9% | 42.1% | **93.9%** |
| CPGCN (Di Jiang & Yang, 2022) | ✓ | – | 26.9% | 6.9% | 93.1% |
| **OTOv2** (80% group sparsity) | ✓ | **14-51-77-122-183-146-92-41-16-13-8-11-14**-107-183 | **23.7%** | **4.9%** | 93.2% |

such as ANNC and SFW-pruning (Miao et al., 2021) do not construct slimmer models besides merely projecting variables onto zero. OTOv2 overcomes all these drawbacks and is the first to realize the end-to-end autonomy for simultaneously training and compressing arbitrary DNNs with high performance. Furthermore, OTOv2 achieves the state-of-the-art results on this intersecting ResNet50 on CIFAR10 experiment. In particular, as shown in Table 3, under 90% group sparsity level, OTOv2 utilizes only 1.2% parameters and 2.2% FLOPs to reach 93.0% top-1 accuracy with slight 0.5% regression. Under 80% group sparsity, OTOv2 achieves competitive 94.5% accuracy to other pruning methods but makes use of substantially fewer parameters and FLOPs.

**ResNet50 on ImageNet.** We finally employ OTOv2 to ResNet50 on ImageNet. Similarly to other experiments, OTOv2 first automatically partitions the trainable variables of ResNet50 into ZIGs (see Figure 11 in Appendix D), and then trains it once by DHSPG to automatically construct slimmer models without fine-tuning. We report a performance portfolio under various target group sparsities ranging from 40% to 70% and compare with other state-of-the-art methods in Figure 6. Remark here that more reliably controlling the ultimate sparsity level to meet various deployment environments is a significant superiority of DHSPG to the HSPG. An increasing target group sparsity results in more FLOPs and parameter quantity reductions, meanwhile sacrifices more accuracy. It is noticeable that OTOv2 roughly exhibits a Pareto frontier in terms of top-1 accuracy and

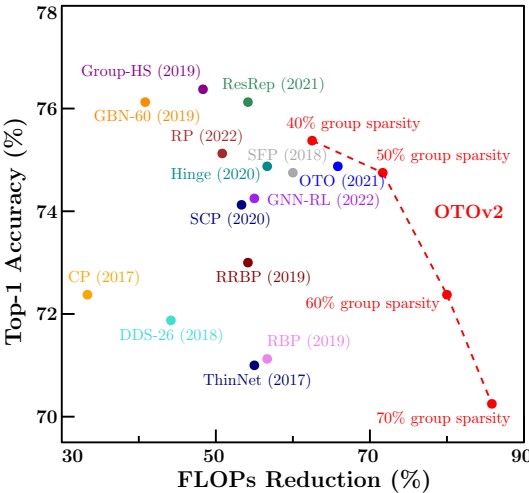

Figure 6: ResNet50 on ImageNet.

FLOPs reduction under various group sparsities. In particular, under 70% group sparsity, the slimmer ResNet50 by OTOv2 achieves fewer FLOPs (14.5%) than others with a 70.3% top-1 accuracy which is competitive to SFP (He et al., 2018a) and RBP (Zhou et al., 2019) especially under 3x fewer FLOPs. The one with 72.3% top-1 accuracy under group sparsity as 60% is competitive to CP (He et al., 2017), DDS-26 (Huang & Wang, 2018) and RRBP (Zhou et al., 2019), but 2-3 times more efficient. The slimmer ResNet50s under 40% and 50% group sparsity achieve the accuracy milestone, *i.e.*, around 75%, both of which FLOPs reductions outperform most of state-of-the-arts. ResRep (Ding et al., 2021), Group-HS (Yang et al., 2019) and GBN-60 (You et al., 2019) achieve over 76% accuracy but consume more FLOPs than OTOv2 and are not automated for general DNNs.

## 5 CONCLUSION

We propose OTOv2 that automatically trains a general DNN only once and compresses it into a more compact counterpart without pre-training or fine-tuning to significantly reduce its FLOPs and parameter quantity. The success stems from two major improvements upon OTOv1: *(i)* automated ZIG partition and automated compressed model construction; and *(ii)* DHSPG method to more reliably solve structured-sparsity problem. We leave the incorporation with NAS as future work.

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

# A   IMPLEMENTATION DETAILS

## A.1   LIBRARY IMPLEMENTATION

The implementation of the current version of OTOv2 (February, 2023) depends on Pytorch (Paszke et al., 2019) and ONNX (Bai et al., 2019) which is an open industrial standard for machine learning interoperability and widely used in numerous AI products of the majority of top-tier industries (see the partners in https://onnx.ai/). In particular, the operators and the connectivity of a general DNN are retrieved by calling the ONNX optimization API of Pytorch, which is the first step to establish the trace graph of OTOv2 in Algorithm 2. The proposed DHSPG is implemented as an instance of the optimizer class for Pytorch. The ultimate compact model construction in Section 3.3 is implemented by modifying the attributes and parameters of the vertices in onnx models according to $x^*_{\text{DHSPG}}$ and ZIGs. As a result, OTOv2 realizes an end-to-end pipeline to automatically and conveniently produce a compact model that meets inference restrictions and can be directly deployed onto product environments. In addition, the constructed compact DNNs in onnx format can be converted back into either torch or tensorflow formats if needed by open-source tools (Arseny, 2022).

## A.2   LIMITATIONS OF BETA VERSION

**Dependency.** The current version of OTOv2 (February, 2023) depends on the ONNX optimization API in Pytorch to obtain vertices (operations) and the connections among them, *i.e.*, $(\mathcal{E}, \mathcal{V})$ in line 2 in Algorithm 2. It is the foremost step for establishing the trace graph of DNN for automated ZIG partition. Therefore, the DNNs that do not comply with this API are not supported by the beta version of OTOv2 yet. We notice that Transformers sometimes have incompatibility issue for its position embedding layers, whereas its trainable part such as encoder layers does not. The current limitation is in the view of engineering perspective and would be resolved following the active and rapid developments of the ONNX and PyTorch community driven by the industry and academy.

**Unknown Operators.** For sanity, we exclude the connected components that possess uncertain/unknown vertices for forming ZIGs in Algorithm 2. This mechanism largely ensures the generality of the automated ZIG partition onto general DNNs. But the ignorance over these connected components may miss some valid ZIGs thereby may leave redundant structures to be unpruned. We will maintain and update the operator list which currently consists of 31 (known/certain) operators to better exploit ZIGs.

## A.3   EXPERIMENTAL DETAILS

We conducted the experiments on one NVIDIA A100 GPU Server. For the experiments in the main body, we estimated the gradient via sampling a mini-batch of data points under first-order momentum with coefficient as 0.9. The mini-batch sizes follow other related works from $\{64, 128, 256\}$. All experiments in the main body share the same commonly-used learning rate scheduler that start from $10^{-1}$ and periodically decay by 10 till $10^{-4}$ every $T_{period}$ epochs. The length of decaying period $T_{period}$ depends on the maximum epoch, *i.e.*, 120 for ImageNet and 300 for others.

In general, we follow the usage of Half-Space projector in (Chen et al., 2021b) to trigger it when the learning rate is first decayed after $T_{period}$ epochs. $\mathcal{G}_p$ and $\mathcal{G}_{np}$ are constructed after $T_{period}/2$ warm up epochs empirically in our experiments. To compute the saliency score, we jointly consider both cosine-similarity and magnitude of each group $g \in \mathcal{G}$. For the groups $g \in \mathcal{G}_p$ which magnitudes need to be penalized, we set $\lambda_g$ in Algorithm 3 as $\lambda_g = \Lambda := 10^{-3}$ if the regularization coefficient does not need to be adjusted, *i.e.*, $\cos(\theta_g) \geq 0$. Note that $\Lambda := 10^{-3}$ is the commonly used coefficient in the sparsity regularization literatures (Chen et al., 2021b; Xiao & Zhang, 2014). Otherwise, we computed the $\lambda_{\min,g} := -\cos(\theta_g) \|[\nabla f(x)]_g\|_2$ and $\lambda_{\max,g} := -\frac{\|[\nabla f(x)]_g\|_2}{\cos(\theta_g)}$ and set $\lambda_g$ by amplifying $\lambda_{\min,g}$ by 1.1 and projecting back to $\lambda_{\max,g}$ if exceeding.

# B   EXTENSIVE ABLATION STUDY

In this appendix, we present additional experiments to demonstrate the superiority of DHSPG over finding local optima with higher performance than HSPG. As described in the main body, the main

advantages of DHSPG compared with HSPG in OTOv1 are *(i)* enlarging search space to be capable of finding local optima with higher performance if any, and *(ii)* more reliably guarantee ultimate group sparsity level. The later one has been demonstrated by the experiments of ResNet50, where DHSPG can precisely achieve different prescribed group sparsity level to meet the requirements of various deploying environments. In contrast, HSPG lacks capacity to achieve a specific group sparsity level due to the implicit relationship between the regularization coefficient $\lambda$ and the sparsity level. The former one will be validated in this appendix. In depth, one takeaway of DHSPG is to separate groups of variables, then treat them via different and specifically designed mechanisms which greatly enlarge the search space. However, HSPG applies the same mechanism to update all variables, which may easily result in convergence to the origin and may be not optimal. In addition, we also provide the runtime comparison in Appendix B.3.

## B.1 SUPER RESOLUTION

We select the popular model architecture CARN (Ahn et al., 2018) for super-resolution task with the scaling factor of two. As (Oh et al., 2022), we use benchmark DIV2K dataset (Agustsson & Timofte, 2017) for training and Set14 (Zeyde et al., 2010), B100 (Martin et al., 2001) and Urban100 (Huang et al., 2015) datasets for evaluation. Similarly to other experiments presented in the main body, OTOv2 automatically partitions the trainable variables of CARN into ZIGs (see Figure 8 in Appendix D). Then we follow the training procedure in (Agustsson & Timofte, 2017) to apply the Adam strategy into DHSPG, *i.e.*, utilizing both first and second order momentums to compute a gradient estimate as line 5 in Algorithm 3. Under the same learning scheduler and total number of steps as the baseline, we conduct both DHSPG and HSPG to compute solutions with high group sparsity, where we set target group sparsity as 50% for DHSPG and fine-tune the regularization coefficient $\lambda$ for HSPG as the power of 10 from $10^{-3}$ to $10^3$ to pick up the one with significant FLOPs and parameters reductions with satisfactory performance. Finally, the more compact CARN models are constructed via the automated compressed model construction in Section 3.3. We report the final results in Table 4.

Table 4: OTOv2 under DHSPG versus HSPG on CARNx2.

| Method | Optimizer | FLOPS | # of Params | PSNR | | |
|---|---|---|---|---|---|---|
| | | | | Set14 | B100 | Urban100 |
| Baseline | Adam | 100% | 100% | 33.5 | 32.1 | 31.5 |
| **OTOv2** | HSPG | 35.5% | 35.4% | 33.0 | 31.6 | 30.9 |
| **OTOv2** | **DHSPG** | **24.3%** | **24.1%** | **33.2** | **31.9** | **31.1** |

Unlike the classification experiments where HSPG and DHSPG perform quite competitively, OTOv2 with DHSPG significantly outperforms OTOv2 with HSPG on this super-resolution task via CARN by using 46% fewer FLOPs and parameters to achieve significantly better PSNR on these benchmark datasets. It exhibits a strong evidence to show the higher generality of DHSPG to enlarge the search space rather than restrict it near the origin point to fit more general applications.

## B.2 BERT ON SQUAD

We next compare DHSPG versus HSPG on pruning the large-scale transformer Bert (Vaswani et al., 2017), evaluated on Squad, a question-answering benchmark (Rajpurkar et al., 2016). Remark here that since Transformer are not reliably compatible with PyTorch's ONNX optimization API at this moment, they can not enjoy the end-to-end autonomy of OTOv2 yet. To compare two optimizers, we apply DHSPG onto the OTOv1 framework, which manually conducts ZIG partition and constructs compressed Bert without fine-tuning. The results are reported in Table 5.

Based on Table 5, it is apparent that DHSPG performs significantly better than HSPG and Prox-SSI (Deleu & Bengio, 2021) by achieving 83.8%-87.7% F1-scores and 74.6%-80.0% exact match rates. In contrast, HSPG and ProxSSI reach 82.0%-84.1% F1-scores and 71.9%-75.0% exact match rates. The underlying reason of such remarkable improvement by DHSPG is that DHSPG enlarges to search space away from trapping around origin points by partitioning groups into magnitude-penalized or not and treating them separately. However, both ProxSSI and HSPG penalize the magnitude of all variables and apply the same update mechanism onto all variables, which deteriorate

the performance significantly in this experiment. The results well validate the effectiveness of the design of DHSPG to enlarge search space for typical better generalization performance.

Table 5: Numerical results of Bert on Squad.

| Method | # of Params | Exact | F1-score | Inference SpeedUp |
|---|---|---|---|---|
| Baseline | 100% | 81.0% | 88.3% | 1× |
| ProxSSI (Deleu & Bengio, 2021) | 83.4%† | 72.3% | 82.0% | 1× |
| OTOv1 + **DHSPG** (10% group sparsity) | 93.3% | **80.0%** | **87.7%** | 1.1× |
| OTOv1 + **DHSPG** (30% group sparsity) | 80.1% | 79.4% | 87.3% | 1.2× |
| OTOv1 + **DHSPG** (50% group sparsity) | 68.3% | 78.1% | 86.2% | 1.3× |
| OTOv1 + **DHSPG** (70% group sparsity) | **55.0%** | 74.6% | 83.8% | **1.4×** |
| OTOv1 + HSPG | 91.0% | 75.0% | 84.1% | 1.1× |
| OTOv1 + HSPG | 66.7% | 71.9% | 82.0% | 1.3× |

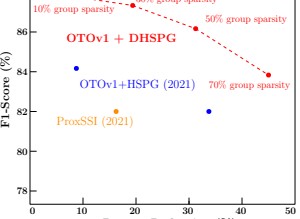

† Approximate value based on the group sparsity reported in (Deleu & Bengio, 2021).

## B.3 RUNTIME COMPARISON

We provide runtime comparison of the proposed DHSPG versus the optimization algorithms used in the benchmark baseline experiments. In particular, we calculate the average runtime per epoch and report the relative runtime in Figure 7. Based on Figure 7, we observe that the per epoch cost of DHSPG is competitive to other standard optimizers. In addition, OTOv2 only trains the DNN once with the similar amount of total epochs for training the baselines. However, other compression methods have to proceed multi-stage training procedures including pretraining the baselines by the standard optimizers, thereby are not training efficient compared to OTOv2.

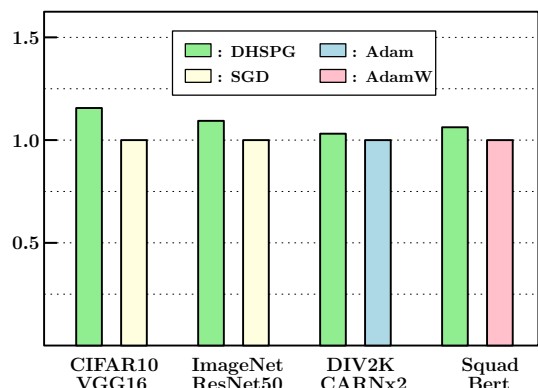

Figure 7: Average runtime per epoch relative comparison.

## C CONVERGENCE ANALYSIS

In this appendix, we provide the rough convergence analysis for the proposed DHSPG. This paper is application-track and mainly focuses on deep learning compression and infrastructure but not the theoretical convergence. Therefore, for simplicity, we assume full gradient estimate at each iteration. More rigorous analysis under stochastic settings will be left as future work,

**Lemma 1.** *The objective function $f$ satisfies*

$$f(\boldsymbol{x}+\alpha\boldsymbol{d}(\boldsymbol{x})) \le f(\boldsymbol{x}) - \left(\alpha - \frac{L\alpha^2}{2}\right)\|\nabla f(\boldsymbol{x})\|_2^2 + \frac{L\alpha^2}{2}\sum_{g\in\mathcal{G}_p}\lambda_g^2 + (L\alpha-1)\alpha\sum_{g\in\mathcal{G}_p}\lambda_g\cos(\theta_g)\|[\nabla f(\boldsymbol{x})]_g\|_2.$$

(4)

*Proof.* By the algorithm,

$$\boldsymbol{d}(\boldsymbol{x}) = \begin{cases} -[\nabla f(\boldsymbol{x})]_g - \lambda_g\frac{[\boldsymbol{x}]_g}{\|[\boldsymbol{x}]_g\|_2} & \text{if } g \in \mathcal{G}_p, \\ -[\nabla f(\boldsymbol{x})]_g & \text{otherwise.} \end{cases}$$

(5)

We can rewrite the direction $[\boldsymbol{d}(\boldsymbol{x})]_g$ for $g \in \mathcal{G}_p$ as the summation of two parts,

$$[\boldsymbol{d}(\boldsymbol{x})]_g = \left[\hat{\boldsymbol{d}}(\boldsymbol{x}) + \tilde{\boldsymbol{d}}(\boldsymbol{x})\right]_g, \tag{6}$$

where

$$\left[\hat{\boldsymbol{d}}(\boldsymbol{x})\right]_g^\top [\nabla f(\boldsymbol{x})]_g = 0, \text{ and } \left\|\left[\hat{\boldsymbol{d}}(\boldsymbol{x})\right]_g\right\| = \left\|-\lambda_g \frac{[\boldsymbol{x}]_g}{\|[\boldsymbol{x}]_g\|} \cdot \cos\left(\theta_g - 90°\right)\right\| = \lambda_g \sin\left(\theta_g\right). \tag{7}$$

Consequently,

$$
\begin{aligned}
\left\|\left[\tilde{\boldsymbol{d}}(\boldsymbol{x})\right]_g\right\|^2 &= \left\|[\boldsymbol{d}(\boldsymbol{x})]_g\right\|^2 - \left\|\left[\hat{\boldsymbol{d}}(\boldsymbol{x})\right]_g\right\|^2 \\
&= \left\|-[\nabla f(\boldsymbol{x})]_g - \lambda \frac{[\boldsymbol{x}]_g}{\|[\boldsymbol{x}]_g\|}\right\|^2 - \lambda_g^2 \sin^2\left(\theta_g\right) \\
&= \left\|[\nabla f(\boldsymbol{x})]_g\right\|^2 + \lambda_g^2 + 2[\nabla f(\boldsymbol{x})]_g^\top \lambda_g \frac{[\boldsymbol{x}]_g}{\|[\boldsymbol{x}]_g\|} - \lambda_g^2 \sin^2\left(\theta_g\right) \\
&= \|[\nabla f(\boldsymbol{x})]_g\|^2 + \lambda_g^2 \cos^2\left(\theta_g\right) + 2\lambda_g \|[\nabla f(\boldsymbol{x})]_g\| \cos\left(\theta_g\right) \\
&= [\|[\nabla f(\boldsymbol{x})]_g\| + \lambda_g \cos\left(\theta_g\right)]^2,
\end{aligned}
\tag{8}
$$

and

$$[\tilde{\boldsymbol{d}}(\boldsymbol{x})]_g = -\frac{\|[\nabla f(\boldsymbol{x})]_g\| + \lambda_g \cos\left(\theta_g\right)}{\|[\nabla f(\boldsymbol{x})]_g\|}[\nabla f(\boldsymbol{x})]_g := -\omega_g[\nabla f(\boldsymbol{x})]_g \tag{9}$$

By the descent lemma, we have that

$$f(\boldsymbol{x} + \alpha\boldsymbol{d}(\boldsymbol{x}))$$

$$\leq f(\boldsymbol{x}) + \alpha\nabla f(\boldsymbol{x})^\top \boldsymbol{d}(\boldsymbol{x}) + \frac{L\alpha^2}{2}\|\boldsymbol{d}(\boldsymbol{x})\|_2^2$$

$$= f(\boldsymbol{x}) + \alpha[\nabla f(\boldsymbol{x})]_{\mathcal{G}_{np}}^\top [\boldsymbol{d}(\boldsymbol{x})]_{\mathcal{G}_{np}} + \alpha[\nabla f(\boldsymbol{x})]_{\mathcal{G}_p}^\top [\boldsymbol{d}(\boldsymbol{x})]_{\mathcal{G}_p} + \frac{L\alpha^2}{2}\left\|[\boldsymbol{d}(\boldsymbol{x})]_{\mathcal{G}_{np}}\right\|_2^2 + \frac{L\alpha^2}{2}\left\|[\boldsymbol{d}(\boldsymbol{x})]_{\mathcal{G}_p}\right\|_2^2$$

$$= f(\boldsymbol{x}) - \left(\alpha - \frac{L\alpha^2}{2}\right)\left\|\nabla f(\boldsymbol{x})]_{\mathcal{G}_{np}}\right\|^2 + \alpha[\nabla f(\boldsymbol{x})]_{\mathcal{G}_p}^\top [\boldsymbol{d}(\boldsymbol{x})]_{\mathcal{G}_p} + \frac{L\alpha^2}{2}\left\|[\boldsymbol{d}(\boldsymbol{x})]_{\mathcal{G}_p}\right\|_2^2$$

$$= f(\boldsymbol{x}) - \left(\alpha - \frac{L\alpha^2}{2}\right)\left\|\nabla f(\boldsymbol{x})]_{\mathcal{G}_{np}}\right\|^2 + \alpha[\nabla f(\boldsymbol{x})]_{\mathcal{G}_p}^\top \left[\hat{\boldsymbol{d}}(\boldsymbol{x}) + \tilde{\boldsymbol{d}}(\boldsymbol{x})\right]_{\mathcal{G}_p} + \frac{L\alpha^2}{2}\left\|\left[\hat{\boldsymbol{d}}(\boldsymbol{x}) + \tilde{\boldsymbol{d}}(\boldsymbol{x})\right]_{\mathcal{G}_p}\right\|_2^2$$

$$= f(\boldsymbol{x}) - \left(\alpha - \frac{L\alpha^2}{2}\right)\left\|\nabla f(\boldsymbol{x})]_{\mathcal{G}_{np}}\right\|^2 + \alpha[\nabla f(\boldsymbol{x})]_{\mathcal{G}_p}^\top \left[\tilde{\boldsymbol{d}}(\boldsymbol{x})\right]_{\mathcal{G}_p} + \frac{L\alpha^2}{2}\left\|\left[\hat{\boldsymbol{d}}(\boldsymbol{x})\right]_{\mathcal{G}_p}\right\|_2^2$$

$$+ L\alpha^2 \left[\hat{\boldsymbol{d}}(\boldsymbol{x})\right]_{\mathcal{G}_p}^\top \left[\tilde{\boldsymbol{d}}(\boldsymbol{x})\right]_{\mathcal{G}_p} + \frac{L\alpha^2}{2}\left\|\left[\tilde{\boldsymbol{d}}(\boldsymbol{x})\right]_{\mathcal{G}_p}\right\|_2^2$$

$$= f(\boldsymbol{x}) - \left(\alpha - \frac{L\alpha^2}{2}\right)\left\|\nabla f(\boldsymbol{x})]_{\mathcal{G}_{np}}\right\|^2 - \alpha\left[\nabla f(\boldsymbol{x})\right]_{\mathcal{G}_p}^\top \sum_{g \in \mathcal{G}_p} \frac{\left\|[\nabla f(\boldsymbol{x})]_g\right\| + \lambda_g \cos\left(\theta_g\right)}{\left\|[\nabla f(\boldsymbol{x})]_g\right\|}[\nabla f(\boldsymbol{x})]_g$$

$$+ \frac{L\alpha^2}{2}\sum_{g \in \mathcal{G}_p} \lambda_g^2 \sin^2\left(\theta_g\right) + \frac{L\alpha^2}{2}\sum_{g \in \mathcal{G}_p} \left(\|[\nabla f(\boldsymbol{x})]_g\|_2 + \lambda_g \cos\left(\theta_g\right)\right)^2$$

$$= f(\boldsymbol{x}) - \left(\alpha - \frac{L\alpha^2}{2}\right)\left\|\nabla f(\boldsymbol{x})]_{\mathcal{G}_{np}}\right\|^2 - \left(\alpha - \frac{L\alpha^2}{2}\right)\left\|[\nabla f(\boldsymbol{x})]_{\mathcal{G}_p}\right\|_2^2 - \alpha\sum_{g \in \mathcal{G}_p} \lambda_g \cos\left(\theta_g\right) \|[\nabla f(\boldsymbol{x})]_g\|$$

$$+ \frac{L\alpha^2}{2}\sum_{g \in \mathcal{G}_p} \lambda_g^2 \left(\sin^2\left(\theta_g\right) + \cos^2\left(\theta_g\right)\right) + L\alpha^2 \sum_{g \in \mathcal{G}_p} \lambda_g \|[\nabla f(\boldsymbol{x})]_g\| \cos\left(\theta_g\right)$$

$$= f(\boldsymbol{x}) - \left(\alpha - \frac{L\alpha^2}{2}\right)\|\nabla f(\boldsymbol{x})\|_2^2 + \frac{L\alpha^2}{2}\sum_{g \in \mathcal{G}_p} \lambda_g^2 + (L\alpha - 1)\alpha \sum_{g \in \mathcal{G}} \lambda_g \cos\left(\theta_g\right) \|[\nabla f(\boldsymbol{x})]_g\|$$

$\square$

**Lemma 2.** *Suppose $\alpha \leq \frac{1}{L}$ and $f$ is L-smooth, then there exists some positive $\lambda_g \in (\lambda_{min,g}, \lambda_{max,g})$ for any $g \in \mathcal{G}_p$ such as*

$$f(\boldsymbol{x} + \alpha \boldsymbol{d}(\boldsymbol{x})) \leq f(\boldsymbol{x}) - \left(\alpha - \frac{L\alpha^2}{2}\right) \left\|[\nabla f(\boldsymbol{x})]_{\mathcal{G}_{np}}\right\|_2^2 \tag{10}$$

*Proof.* Based on Lemma 1 and $\alpha \leq \frac{1}{L}$, we have that

$$f(\boldsymbol{x} + \alpha \boldsymbol{d}(\boldsymbol{x}))$$

$$\leq f(\boldsymbol{x}) - \left(\alpha - \frac{L\alpha^2}{2}\right) \|\nabla f(\boldsymbol{x})\|_2^2 + \frac{L\alpha^2}{2} \sum_{g \in \mathcal{G}_p} \lambda_g^2 + (L\alpha - 1)\alpha \sum_{g \in \mathcal{G}} \lambda_g \cos(\theta_g) \left\|[\nabla f(\boldsymbol{x})]_g\right\|_2,$$

$$\leq f(\boldsymbol{x}) - \left(\alpha - \frac{L\alpha^2}{2}\right) \left\|[\nabla f(\boldsymbol{x})]_{\mathcal{G}_{np}}\right\|_2^2 + \sum_{g \in \mathcal{G}_p} h(\lambda_g, g)$$

$$\tag{11}$$

where we denote

$$h(\lambda_g, g) := \frac{L\alpha^2 \lambda_g^2}{2} + (L\alpha - 1)\alpha \cos(\theta_g) \left\|[\nabla f(\boldsymbol{x})]_g\right\|_2 \lambda_g - \left(\alpha - \frac{L\alpha^2}{2}\right) \left\|[\nabla f(\boldsymbol{x})]_g\right\|_2^2. \tag{12}$$

We can see that for any $g \in \mathcal{G}_p$, then $h(\lambda, g) \leq 0$ if and only if the following holds

$$\lambda_g \leq \underbrace{\frac{(1 - L\alpha)\alpha \cos(\theta_g) \left\|[\nabla f(\boldsymbol{x})]_g\right\|_2 + \sqrt{(1 - L\alpha)^2 \alpha^2 \cos^2(\theta_g) \left\|[\nabla f(\boldsymbol{x})]_g\right\|_2^2 + 2L\alpha^2 \left(\alpha - \frac{L\alpha^2}{2}\right) \left\|[\nabla f(\boldsymbol{x})]_g\right\|_2^2}}{L\alpha^2}}_{:=\hat{\lambda}_g}.$$

$$\tag{13}$$

Next, we need to show for the group $g$ that requires $\lambda_g$ adjustment, the $(\lambda_{min,g}, \hat{\lambda}_g)$ is a valid interval, *i.e.*, $\hat{\lambda}_g \geq \lambda_{min,g}$. To show it, combining with $\lambda_{min,g} = -\cos(\theta_g) \left\|[\nabla f(\boldsymbol{x})]_g\right\|_2$, we have that

$$\hat{\lambda}_g = \frac{(L\alpha - 1)\alpha \lambda_{min,g} + \sqrt{(1 - L\alpha)^2 \alpha^2 \lambda_{min,g}^2 + 2L\alpha^2 \left(\alpha - \frac{L\alpha^2}{2}\right) \lambda_{min,g}^2 / \cos^2(\theta_g)}}{L\alpha^2}$$

$$= \frac{(L\alpha - 1)\lambda_{min,g} + \lambda_{min,g}\sqrt{1 - L^2\alpha^2 \tan^2(\theta_g) + 2L\alpha \tan^2(\theta_g)}}{L\alpha}$$

$$= \frac{(L\alpha - 1)\lambda_{min,g} + \lambda_{min,g}\sqrt{-(L\alpha - 1)^2 \tan^2(\theta_g) + \tan^2(\theta_g) + 1}}{L\alpha} \tag{14}$$

$$> \frac{(L\alpha - 1)\lambda_{min,g} + \lambda_{min,g}}{L\alpha}$$

$$= \lambda_{min,g},$$

where the second last inequality holds since $0 < \alpha \leq 1/L$, then

$$\sqrt{-(L\alpha - 1)^2 \tan^2(\theta_g) + \tan^2(\theta_g) + 1} > 1.$$

Then, we need to show that $\hat{\lambda}_g$ is greater than 0. There are two cases to be considered:

- $\cos\theta_g < 0$: then $\hat{\lambda}_g \geq \lambda_{min,g} = -\cos(\theta_g) \left\|[\nabla f(\boldsymbol{x})]_g\right\|_2 > 0$.

- $\cos\theta_g \geq 0$: it follows $0 < \alpha < 1/L$ and (13) that $\hat{\lambda}_g > 0$.

Thus for $\lambda_g \in (\lambda_{min,g}, \min\{\lambda_{max,g}, \hat{\lambda}_g\})$, $h(\lambda_g, g) \leq 0$. Consequently, there exists some positive $\lambda_g \in (\lambda_{min,g}, \lambda_{max,g})$ so that $h(\lambda_g, g) \leq 0$. Finally, the proof is complete if we choose $\lambda_g$ satisfies the above for any $g \in \mathcal{G}_p$,

$$f(\boldsymbol{x} + \alpha \boldsymbol{d}(\boldsymbol{x})) \leq f(\boldsymbol{x}) - \left(\alpha - \frac{L\alpha^2}{2}\right) \left\|[\nabla f(\boldsymbol{x})]_{\mathcal{G}_{np}}\right\|_2^2 + \sum_{g \in \mathcal{G}_p} h(\lambda, g)$$

$$\leq f(\boldsymbol{x}) - \left(\alpha - \frac{L\alpha^2}{2}\right) \left\|[\nabla f(\boldsymbol{x})]_{\mathcal{G}_{np}}\right\|_2^2. \tag{15}$$

$\square$

**Lemma 3.** *For any $g \in \mathcal{G}_p$, if $0 < \alpha < \frac{2[\boldsymbol{x}]_g^\top [\boldsymbol{d}(\boldsymbol{x})]_g}{\|[\boldsymbol{d}(\boldsymbol{x})]_g\|_2^2}$, then the magnitude of the variables satisfies*

$$\left\| [\boldsymbol{x} + \alpha \boldsymbol{d}(\boldsymbol{x})]_g \right\|_2 < \left\| [\boldsymbol{x}]_g \right\|_2 . \tag{16}$$

*And if $\alpha = \omega \frac{[\boldsymbol{x}]_g^\top [\boldsymbol{d}(\boldsymbol{x})]_g}{\|[\boldsymbol{d}(\boldsymbol{x})]_g\|_2^2}$ for $\omega \in (0,1)$, then*

$$\left\| [\boldsymbol{x} + \alpha \boldsymbol{d}(\boldsymbol{x})]_g \right\|_2^2 = \left\| [\boldsymbol{x}]_g \right\|_2^2 + (\omega^2 - 2\omega) \left\| [\boldsymbol{x}]_g \right\|_2^2 \cos^2(\theta_g). \tag{17}$$

*Proof.*

$$
\begin{aligned}
&\left\| [\boldsymbol{x} + \alpha \boldsymbol{d}(\boldsymbol{x})]_g \right\|_2^2 \\
&= \left\| [\boldsymbol{x}]_g - \alpha \left( [\nabla f(\boldsymbol{x})]_g + \lambda_g \frac{[\boldsymbol{x}]_g}{\|[\boldsymbol{x}]_g\|_2} \right) \right\|_2^2 \\
&= \| [\boldsymbol{x}]_g \|_2^2 - 2\alpha [\boldsymbol{x}]_g^\top \left( [\nabla f(\boldsymbol{x})]_g + \lambda_g \frac{[\boldsymbol{x}]_g}{\|[\boldsymbol{x}]_g\|_2} \right) + \alpha^2 \left\| [\nabla f(\boldsymbol{x})]_g + \lambda_g \frac{[\boldsymbol{x}]_g}{\|[\boldsymbol{x}]_g\|_2} \right\|_2^2 \\
&= \| [\boldsymbol{x}]_g \|_2^2 + t(\alpha),
\end{aligned} \tag{18}
$$

where

$$t(\alpha) = -2\alpha [\boldsymbol{x}]_g^\top \left( [\nabla f(\boldsymbol{x})]_g + \lambda_g \frac{[\boldsymbol{x}]_g}{\|[\boldsymbol{x}]_g\|_2} \right) + \alpha^2 \left\| [\nabla f(\boldsymbol{x})]_g + \lambda_g \frac{[\boldsymbol{x}]_g}{\|[\boldsymbol{x}]_g\|_2} \right\|_2^2 = A\alpha^2 - 2B\alpha \tag{19}$$

and

$$A := \left\| [\nabla f(\boldsymbol{x})]_g + \lambda_g \frac{[\boldsymbol{x}]_g}{\|[\boldsymbol{x}]_g\|_2} \right\|_2^2 > 0 \tag{20}$$

$$B := [\boldsymbol{x}]_g^\top \left( [\nabla f(\boldsymbol{x})]_g + \lambda_g \frac{[\boldsymbol{x}]_g}{\|[\boldsymbol{x}]_g\|_2} \right) > 0. \tag{21}$$

note that $B > 0$ because of the selection of $\lambda_g$.

Consequently, we have that if $0 < \alpha < \frac{2B}{A} = \frac{2[\boldsymbol{x}]_g^\top [\boldsymbol{d}(\boldsymbol{x})]_g}{\|[\boldsymbol{d}(\boldsymbol{x})]_g\|_2^2}$, then $t(\alpha) < 0$.

Finally, if $\alpha = \omega \frac{[\boldsymbol{x}]_g^\top [\boldsymbol{d}(\boldsymbol{x})]_g}{\|[\boldsymbol{d}(\boldsymbol{x})]_g\|_2^2} = \omega \frac{B}{A}$ for $\omega \in (0,2)$, then

$$t(\alpha) = A\omega^2 \frac{B^2}{A^2} - 2B\omega \frac{B}{A} = (\omega^2 - 2\omega) \frac{B^2}{A} = (\omega^2 - 2\omega) \| [\boldsymbol{x}]_g \|_2^2 \cos^2(\theta_g), \tag{22}$$

which completes the proof.

$\square$

**Corollary 1.** *Suppose $\alpha = \omega \min_{g \in \mathcal{G}_p} \frac{[\boldsymbol{x}]_g^\top [\boldsymbol{d}(\boldsymbol{x})]_g}{\|[\boldsymbol{d}(\boldsymbol{x})]_g\|_2^2}$ for some $\omega \in (0,1)$ and $|\cos(\theta_g)| \geq \rho$ for $g \in \mathcal{G}_p \cap \mathcal{G}^{\neq 0}(\boldsymbol{x})$ and some positive $\rho \in (0,1]$. Then there exists $\gamma \in (0,1)$ such that*

$$\left\| [\boldsymbol{x} + \alpha \boldsymbol{d}(\boldsymbol{x})]_{\mathcal{G}_p} \right\|_2^2 \leq (1 - \gamma^2) \left\| [\boldsymbol{x}]_{\mathcal{G}_p} \right\|_2^2 . \tag{23}$$

*Proof.* The result can be complete via summing (17) over $\mathcal{G}_p$ and combining with $\alpha$ selection. $\square$

## D   ZIG ILLUSTRATION

In this appendix, for more intuitive illustration, we provide the visualizations of the connected components of dependency for the experimented DNNs throughout the whole paper. They are constructed by performing Algorithm 2 to automatically partition ZIGs. Due to the large-scale and intricacy of graphs, we recommend to *zoom in* for reading greater details (under 200%-1600% zoom-in scale via Adobe PDF reader). The vertices marked as the same color represent one connected component of dependency. See the figures starting from the *next* pages.

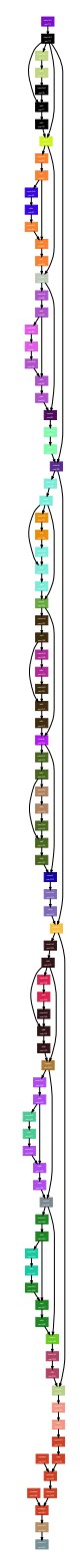

Figure 8: CARN.

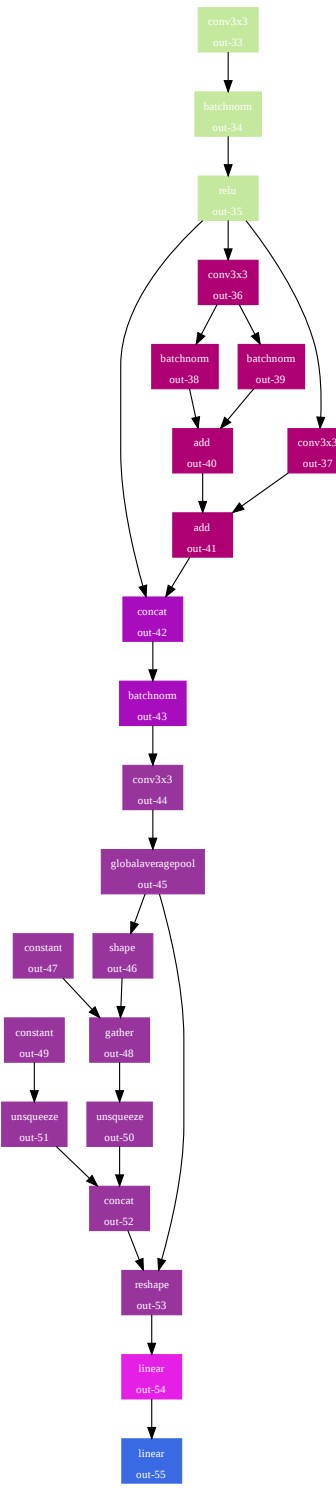

Figure 9: DemoNet.

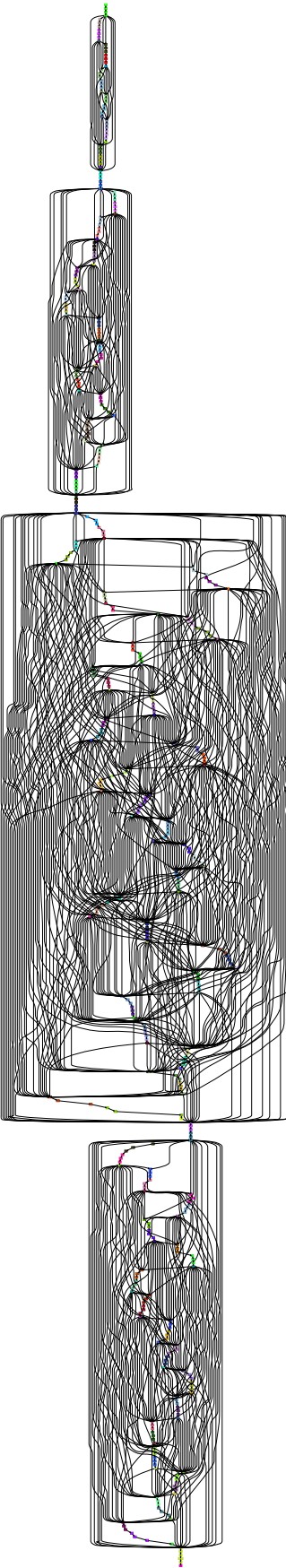

Figure 10: DenseNet121.

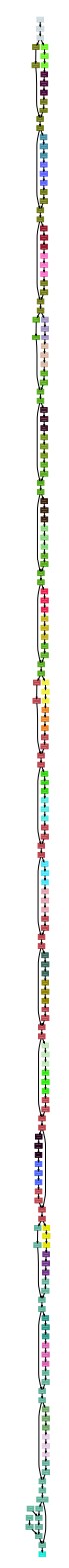

Figure 11: ResNet50.

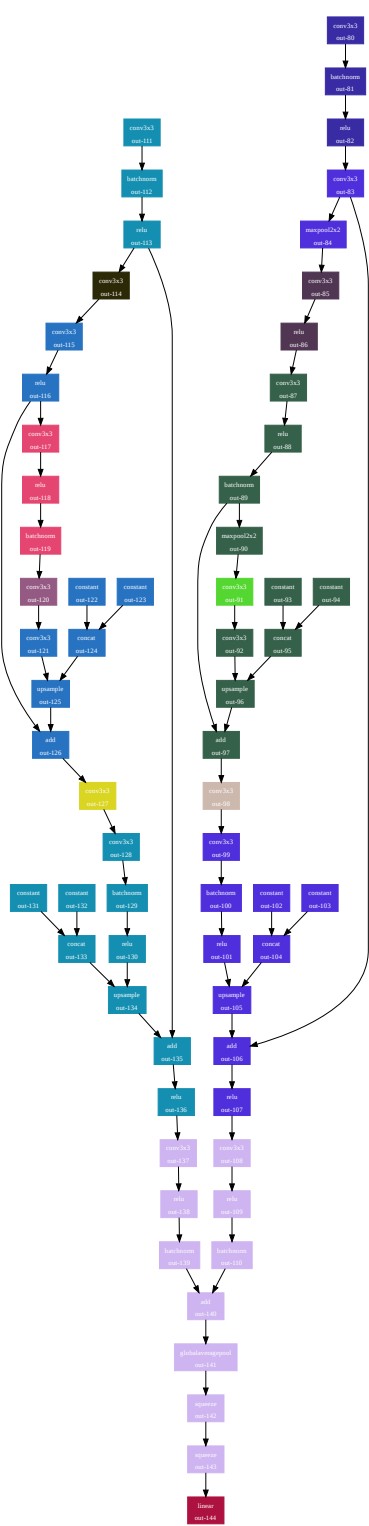

Figure 12: StackedUnets.

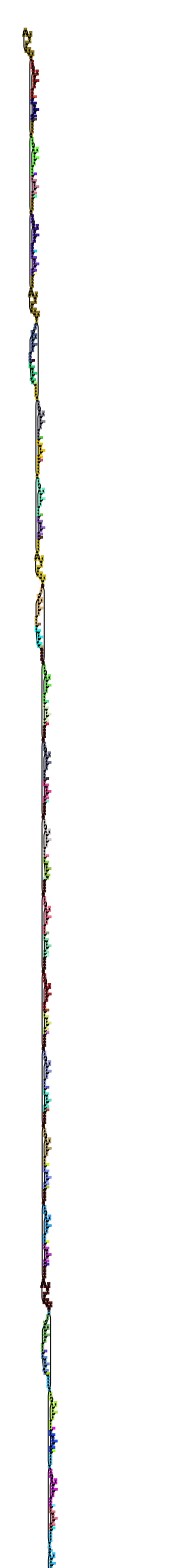

Figure 13: ConvNeXt-Tiny.

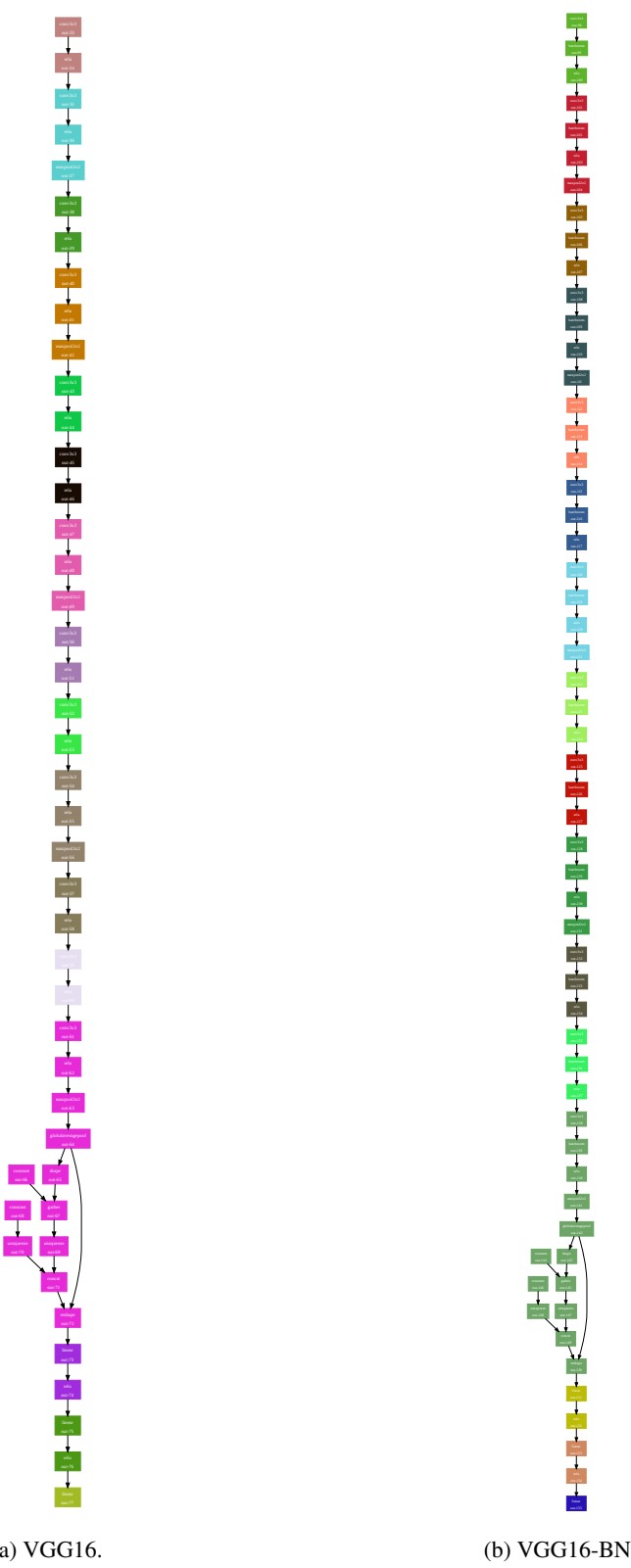

(a) VGG16.

(b) VGG16-BN.

Figure 14: VGG16 and VGG16-BN.

