# OpenReview forum: "OTOv2: Automatic, Generic, User-Friendly"
_ICLR.cc/2023/Conference — ICLR 2023 poster_

### Official Review · Reviewer_eYjt · 2022-10-23

**Confidence:** 2
**Correctness:** 3
**Technical Novelty And Significance:** 2
**Empirical Novelty And Significance:** 3
**Recommendation:** 6

**Clarity, Quality, Novelty And Reproducibility:**

I think the writing of this manuscript can be improved. Regarding the Automatic ZIG partition, it seems like this part is for dependency tracking. What is the core technical contribution in this part? Regarding DHSPG, it is not clear to me why it works. Adding more discussions and illustrations may be helpful.

**Strength And Weaknesses:**

Strength:
1. Compared with the multi-stage model compression procedures, the proposed framework is simple and easy to use.
2. OTOv2 shows better performances than previous methods on several small image classification datasets and ImageNet.

Weakness:
1. Lack of ablation study experiments. It is unclear to me how many improvements the proposed DHSPG bring. Adding ablation study experiments on ImageNet with different model architectures and target sparsities will help resolve this concern.
2. It is unclear whether the proposed method is effective on recent state-of-the-art models. For example, [1] shows that ResNet50 can achieve 80+ ImageNet top1 accuracy with an improved training setting. Adding additional results on these recent state-of-the-art models will help resolve this concern.

[1] ResNet strikes back: An improved training procedure in timm

**Summary Of The Paper:**

This paper presents OTOv2, an improved version of Only-Train-Once (OTOv1) framework. It introduces two major improvements over OTOv1, including automated zero-invariant groups (ZIGs) partition and a new optimizer called Dual Half-Space Projected Gradient
(DHSPG). Experiments are conducted on several small image classification datasets and ImageNet.

**Summary Of The Review:**

comments above.

---

> ### Author Response · Authors · 2022-11-17
> **Revisions and more comparisons are available.**
>
> Dear Reviewer eYjt,
>
> We appreciate your valued questions and address them as below. To address your concerns, we conducted more experiments and included them in the revisions. Please see our responses as below.
>
> - **Lack ablation study of DHSPG.**
>
> Besides the experiments shown in the main body, we presented an ablation study between DHSPG versus HSPG on a benchmark super-resolution task in Appendix B.1. In the revision, we provided two more ablation studies over Bert on Squad in Appendix B.2 and ConvNeXt on ImageNet in Appendix B.3.  All these experiments validate the two major advantages of DHSPG over HSPG, (i) more reliably control ultimate group sparsity level with fewer hyper-parameter tuning, and (ii) enlarge the search space for typically better generalization performance.
>
>
> - **Does OTOv2 work with modernized training tricks in TIMM?**
>
> Thanks for the question. OTOv2 works well and compatibly with these modernized training tricks such as the Mixup and CutMix data augmentations in TIMM. In fact, OTOv2 and these tricks do not conflict but mutually benefit. To demonstrate it and combine with Reviewer F5Jh's suggestions, we conducted a ConvNeXt on ImageNet experiment in the revision, which baseline counterpart was trained with TIMM. The numerical experiments showed that OTOv2 performed normally and reached Top-1 Accuracy over 80% with significant FLOPs reduction.
>
> - **More illustrations regarding DHSPG and automated ZIG partition.**
>
> Thanks for the suggestion. We added more descriptive illustrations to explain the automated ZIG partition and DHSPG in the revision.
>
> - **What is the technical contribution of automated ZIG partition?**
>
> Thanks for the question. The reviewer is correct that tracking dependency is one of the main tasks. But tracking dependency from the raw trace graph is challenging due to complicated connectivities and different roles of the vertices. To address it, we propose a novel automated ZIG partition algorithm, which serves as a fundamental component for the autonomy of OTOv2 to be applied onto general DNN's training and pruning. Its implementation is non-trivial and challenging in the view of both algorithm designs and engineering developments, and can further benefit the pruning methods in the future.
>
> With greater details, our goal is to establish an automated DNN training and pruning framework that given a general DNN, automatically trains it only once into a slimmer architecture with significant FLOPs and params reductions without fine-tuning. To reach it, several key problems need to be systematically resolved. One of them is how to find the minimal groups of parameters across the layers that need to be pruned together, referred to ZIGs. So that after removing ZIGs, the remaining DNN is still valid. Unlike the NAS which remove branches of layers entirely, pruning methods focus on removing the inherent redundancy inside the layers without changing the layer-wise connectivities.  However, the inherent redundant groups across the layers are typically complicated because of the intricacy of DNN architectures. In the past, existing works have to manually find out these groups, which is time-consuming and requires extensive domain-knowledge and engineering efforts. As a result, they are not easily to be applied onto extensive scenarios.
>
> OTOv2 is perhaps the first automated and user-friendly framework that realizes the end-to-end automated general DNN training and pruning procedure. Its effectiveness and autonomy highly relies on our proposed automated ZIG partition, which can further benefit the pruning methods in the future to avoid the time-consuming grouping stage. Finally, automated ZIG partition is challenging in the view of both algorithm designs and infrastructure developments due to the complicated DNN connectivities and the lack of sufficient public APIs.
>
> Sincerely,
>
> Paper 708 Authors.

---

### Official Review · Reviewer_iQEU · 2022-10-25

**Confidence:** 3
**Correctness:** 3
**Technical Novelty And Significance:** 3
**Empirical Novelty And Significance:** 4
**Recommendation:** 6

**Clarity, Quality, Novelty And Reproducibility:**

The novelty and technical depth of the work are both high in my opinion. I did however struggle to understand the paper at times. In particular, I had to re-read the first few sections of the paper repeatedly to grasp what a zero invariant group is. I also found it challenging to parse how the proposed DHSP technique improves over HSPG. I think the paper could be greatly improved if the authors emphasized and clarified these elements of the paper early on.

In places I found the writing to be sensational, which I think took away from the authors' work. For example, the use of "revolutionary" in the abstract, the sentence "Consequently, in both academy and industry, compressing full DNNs into slimmer ones with negligible performance regression becomes ubiquitous.", and the phrase "“guaranteeing ultimate sparsity”.

**Strength And Weaknesses:**

Strengths
- The authors empirical evaluation considers many different tasks and many previously published works.
- The automated detection of constraints that must be respected in order to maintain a valid DNN architecture after group pruning is impactful for lower the cost applying neuron-pruning methods to DNNs.

Weaknesses
- I found it hard to assess the results of experiments in section 4. For each experiment the authors include many prior results in tables. Each technique offers a different tradeoff in terms of accuracy lost to achieve a given level of compression. I believe it would be much easier to understand the results of these two-dimensional comparisons if they were shown in a scatter plot. If OTOv2 is more effective, I would expect it to represent the Pareto frontier of the solutions across a range of compression levels. I am not sure that the reported data is sufficient to demonstrate this. For example, the results in Table 4 show OTOv2 achieving a maximum top-1 accuracy of 75.4%, which is as much as 1% off the highest accuracy reported in the table. I think the experiments could be made to be more effective if the authors focused on thorough analysis of one or two benchmarks rather than incomplete analysis of many.

**Summary Of The Paper:**

The authors propose a set of methods for inducing group sparsity in DNN architecture. Their contributions build on an existing system Only-Train-Once (OTO) by automating the identification of groups that must be pruned together (referred to as zero-invariant groups) and improving on the optimization method, half-space projected gradient descent (HSPG),  by splitting the parameter groups into penalized and non-penalized sets and tuning per-group regularization coefficients.

**Summary Of The Review:**

Overall, I would like to see the authors improve the clarity of the text and the comparisons between their technique and prior work.

---

> ### Author Response · Authors · 2022-11-17
> **A revision incorporating the valued suggestions has been uploaded.**
>
> Dear Reviewer iQEU,
>
> We appreciate your valued suggestions and constructive comments. We provided a revision (with the revised area marked as blue) that hopefully tackled all your suggestions properly. Please see our responses as follows.
>
>
> - **More effective numerical comparison via scatter plot.**
>
> Thank you for the suggestion. We drew a scatter plot over FLOPs reduction versus Top-1 Accuracy and placed it in the revision as Figure 6. We do agree that such scatter plot is indeed more effective for numerical comparison than the previous table. Based on Figure 6, we observed that OTOv2 roughly exhibited the Pareto frontier. Several methods achieved 76%+ Top-1 Accuracy but required more FLOPs and are not automated for general DNNs. The discussion was revised in accordance on page 9.
>
> - **More early-on illustrations regarding ZIG and DHSPG.**
>
> Thanks for the suggestion. We added more descriptive languages and illustrations to emphasize and clarify the two fundamental concepts/components (automated ZIG partition in Section 3.1 and DHSPG in Section 3.2) in the revision. We also conducted more experiments over more architectures, datasets and training tricks, and showed the results in Table 1 and Appendix B in the revision for better illustrating the advantages of proposed framework.
>
>
> - **Writing to be sensational in some places.**
>
> We apologize for these vocabularies and phrases. We carefully revisited the manuscript and polished the writing to make it more proper and precise. Please let us know if any place is still sensational.
>
>
> Sincerely,
>
> Paper 708 Authors.

---

> > ### Comment · Reviewer_iQEU · 2022-11-29
> > **Reviewer Response**
> >
> > Thank you for your revisions. The scatter plot is a large improvement and I appreciate the changes to the text. Based on these changes I've increased my score.
> >
> > I think the paper could be further improved by additional results in the scatter plot below 40% group sparsity. There appears to be a density of techniques with results below this range and it would be useful to see how OTOv2 compares in this regime.

---

### Official Review · Reviewer_F5Jh · 2022-10-25

**Confidence:** 2
**Clarity, Quality, Novelty And Reproducibility:** This work is well-motivated and makes…
**Correctness:** 3
**Technical Novelty And Significance:** 4
**Empirical Novelty And Significance:** 3
**Recommendation:** 8

**Strength And Weaknesses:**

Strength:
- This paper built a model compression system to minimize the human effort put into weight pruning.
- The technical details are well-explained.
- The proposed DHSPG algorithm can satisfy different sparsity requirement without finetuning.
- The empirical results are better or on par with sota pruning works.

Weakness:
- The experiments are still limited to popular standard networks including VGG/ResNet/DenseNet, which have been already very well-studied and known to have large redundancy. To support the claim that the proposed system can work on arbitrary DNN, it would be interesting to see how it does on modern architectures like RegNet/Swin/ConvNeXt.
- It would be great to show some training speed benchmarks.

**Summary Of The Paper:**

In this paper, the authors proposed OTOv2, which is a follow-up work of OTOv1. OTOv2 makes structured pruning more automatic, generic and user-friendly by addressing several problems in OTOv1. Specifically, they proposed an algorithm to automatically find Zero-Invariant Group for arbitrary DNNs and an optimizater to address the sparse optimization problem. Their empirical results is on par with other sota pruning methods, but requires less human efforts.

**Summary Of The Review:**

In conclusion, this paper designs a model pruning system OTOv2. OTOv2 can do ZIG partition automatically for arbitrary neural architectures, optimize the sparse model using a novel optimization algorithm DHSPG and produce the pruned model without finetuning. In my opinion, this work makes clear contribution to the model compression community.

---

> ### Author Response · Authors · 2022-11-17
> **A revision incorporating the valued suggestions is available.**
>
> Dear Reviewer F5Jh,
>
> We appreciate your valued comments and favorable recommendations for our work. We provided a revision that incorporates your suggestions outlined as below.
>
>
> * **More experiments on RegNet, ConvNeXt and Transformer.**
>
> Thanks for the suggestions. We included extensive experiments in the revision.
>
> **RegNet and ConvNeXt.** We conducted RegNet on CIFAR10, ConvNeXt on ImageNet experiments. These architectures contain a few  grouped convolution layers which we adjusted our operator list correspondingly as a variant of standard conv layer. The results in Table 1 of revision show that OTOv2 works well for these modernized model architectures to automatically partition ZIGs, train by DHSPG, then automatically construct slimmer model.
>
> **Transformer.** To study with HSPG in ablation, we conducted a Bert on Squad experiment in Appendix B.2 to significantly speed up Berts with competitive performance to the full models. The results also well validate the effectiveness of the proposed DHSPG in OTOv2 than the HSPG in OTOv1.
>
> Checkpoints have been uploaded onto https://tinyurl.com/checkpointsiclr708.
>
> Furthermore, we added Appendix A.2 to explicitly describe the limitations of current OTOv2 framework and rephrased arbitrary DNNs as general DNNs. In general, as presented in Appendix A.1, OTOv2 depends on PyTorch's ONNX optimization API, i.e., torch.onnx._optimize_trace(), to obtain the DNN's vertices and their connections, which is the first step for automated ZIG partition, i.e., obtain ($\mathcal{E}, \mathcal{V}$) in line 2 of Algorithm 2. Therefore, the DNNs that currently are not supported by this API can not enjoy the end-to-end autonomy of OTOv2 yet. But following the rapid and active developments of the ONNX and PyTorch community, we believe the coverage of OTOv2 will be growing accordingly.
>
> * **Runtime comparison.**
>
> Thanks for the suggestion. We compared the average epoch cost of DHSPG versus the vanilla SGD, Adam and AdamW for training the baseline models as Figure 7 in Appendix B.4, which shows the computational cost of DHSPG is competitive to those of standard optimizers. In addition, OTOv2 trains the DNNs once with the similar amount of total epochs for training the baselines. Meanwhile, other compression methods perform multi-stage training procedures, thereby are typically less efficient than OTOv2.
>
> Sincerely,
>
> Paper 708 Authors

---

### Author Response · Authors · 2022-11-20
**Summary of takeaways and updates from the authors. Look forward to further discussion.**

Dear reviewers and ACs,

Thank you very much again for all your constructive reviews that helped us improve our manuscript. We are glad to see that the novelty, technical depth and practical impact of our proposed one-shot automated model compression framework OTOv2 have been recognized by most reviewers. During awaiting for the finalized review comments, we would like to summarize the key points for a quick understanding of all.

- **Motivations of OTOv2.**

Existing compression methods typically require extensive domain-knowledge and efforts to conduct model-specific compressions, which prevent their broader usages. Meanwhile, they usually require multiple training stages which might be time-consuming. To resolve the pain points, we aim at proposing an automated, generic and user-friendly one-shot DNN training and pruning framework to fit general DNNs.

- **Practical advantages of OTOv2.**

To the best of our knowledge, OTOv2 is the first DNN training and pruning framework that can *automatically* train *general* DNNs from scratch only once and compress them into slimmer counterparts with significant FLOPs and parameters reductions. OTOv2 realizes the end-to-end autonomy from full to slimmer models and requires few engineering efforts from the users. The practical advantages of OTOv2 than others become more significant if the DNN complicatedness is increasing, where other methods are not easily employed without spending sufficient efforts.

- **Two main target problems.**

Realizing end-to-end autonomy for general DNN compression is challenging and needs to systematically address a few key problems. Two of them are (i) How to automatically find out the structures that can be removed?, and (ii) How to identify redundant structures without sacrificing performance? We resolve them by the following key components.


- **Automated ZIG Partition.**

OTOv2 designs and develops an novel algorithm to automatically partition DNN's trainable variables into ZIGs. ZIG (proposed in OTOv1) describes a class of minimal removal structures such that the remaining DNN after removing ZIGs is still valid. Suggested by Reviewer iQEU, ZIG can be largely interpreted as a minimal group of variables that must be pruned together.

Automated ZIG partition algorithm is a series of customized graph algorithms dedicately composed together. Its design and implementaion is challenging due to the complicated connections in DNN's graph, the different roles of vertices, and the lack of sufficient public APIs.

- **DHSPG versus HSPG and proximal methods.**

We propose DHSPG to solve structured sparsity optimization problem and apply it over the ZIGs to find out redundant structures to be further removed.

DHSPG outperforms HSPG and proximal methods to (i) more reliably control group sparsity level, and (ii) enlarge search space for typically better generalization performance. To validate the later one, we highlight Bert on Squad experiment in Appendix B.2, where DHSPG significantly outperforms HSPG and ProxSSI.


**Table. Pruned Berts on SQuAD.**

|Method| # of Params |    Exact    |   F1-Score  |
|------| ----------- | ----------- | ----------- |
|ProxSSI| 83.4% | 72.3% | 82.0% |
| DHSPG (30% group sparsity)| 80.1%|  **79.4%** | **87.3%** |
| DHSPG (70% group sparsity)| **55.0%**|  74.6% | 83.8% |
| HSPG |    91.0%    |    75.0%    |    84.1%    |
| HSPG |    66.7%    |    71.9%    |    82.0%    |

Such performance gap is due to the discrepancy of search spaces.

In fact, HSPG and proximal methods induce group sparsity by solving a non-constrained problem

$minimize_{x} f(x)+\lambda \sum_{g\in G}||[x]_g||_2$.

They have limitations such as hard to control sparsity level since the relation between $\lambda$ and group sparsity is implicit, thereby require time-consuming hyper-parameter fine-tuning. In addition, they generate group sparsity by (i) penalizing the magnitude of all variables to force them near the origin, (ii) then applying proximal or half-space operator to project variables onto zeros. Therefore, such methods restrict the convergence surrounding the origin.

However, the best local minimizers may locate variably upon applications, some of which may stay away from the origin, while HSPG and Proximal methods lack capacity to find them. DHSPG effectively resolves such limitations. In short, it forms a constrained optimization problem and separates ZIGs $G$ into different sets $G_{np}\cup G_p$. Different updating strategies are then applied, i.e., (i) penalizing magnitude with group-wise  $\lambda_g$ and yielding group sparsity over $G_p$ , and (ii) employing SGD or its variants over $G_{np}$ to achieve high performance.

As a result,  DHSPG simultaneously enlarges the search space for better performance and reliably explores group sparsity.

- **Revision.**

In the revision, we included the above discussions and added more experiments and illustrations to address review comments.


Sincerely,

Paper 708 Authors.

---

### Decision · Program_Chairs · 2023-01-20

**Decision:**

Accept: poster

**Justification For Why Not Higher Score:**

I think this paper represents a meaningful step forward, but does not constitute a major intellectual step forward. I expect OTOv2 to be more useful than OTO, but it likely won't transform the field. It's also mainly of interest to a pretty specialized audience.

**Justification For Why Not Lower Score:**

Reviewers all agreed on acceptance.

**Metareview: Summary, Strengths And Weaknesses:**

This paper proposes a follow-up to "only train once" (OTO), a method for model compression. The main innovation of the proposed OTOv2 method is a significant simplification of the model compression pipeline that aims to make model compression much easier for end-users. Reviewers all agreed that this was a valuable contribute and should be accepted.

**Note From Pc:**

if the above contains the word "oral" or "spotlight" please see: "oral" presentation means -> notable-top-5% and "spotlight" means -> notable-top-25%. As stated in our emails, we are disassociating presentation type from AC recommendations